# Estimation of the susceptibility of a road network to shallow landslides with the integration of the sediment connectivity

Massimiliano Bordoni[1], M. Giuseppina Persichillo[1], Claudia Meisina[1], Stefano Crema[2], Marco Cavalli[2], Carlotta Bartelletti[3], Yuri Galanti[3], Michele Barsanti[4], Roberto Giannecchini[3], Giacomo D'Amato Avanzi[3]

[1]Department of Earth and Environmental Sciences, University of Pavia, Pavia, 27100, Italy
[2]Research Institute for Geo-Hydrological Protection, National Research Council, Padova, 35127, Italy
[3]Department of Earth Sciences, University of Pisa, Pisa, 56126, Italy
[4]Department of Civil and Industrial Engineering, University of Pisa, Pisa, 56126, Italy

*Correspondence to*: Massimiliano Bordoni (massimiliano.bordoni01@universitadipavia.it)

**Abstract.** Landslides cause severe damage to road network of a hit zone, in terms of both direct (partial or complete destruction of a road trait, blockages) and indirect (traffic restriction, cut-off of a certain area) costs. Thus, the identification of the parts of the road network which are more susceptible to landslides is fundamental to reduce the risk to the population potentially exposed and the money expense caused by road damaging. For these reasons, this paper aimed to develop and test a data-driven model for the identification of road sectors that are susceptible to be hit by shallow landslides triggered in slopes upstream to the infrastructure. This model was based on the Generalized Additive Method, where the function relating predictors and response variable is an empirically fitted smooth function that allows fitting the data in the more likely functional form, considering also non-linear relations. This work also analyzed the importance, on the estimation of the susceptibility, of considering or not the sediment connectivity, which influences the path and the travel distance of the materials mobilized by a slope failure till a potential barrier as a road. The study was carried out in a catchment of north-eastern OltrepòPavese (northern Italy), where several shallow landslides affected roads in the last 8 years. The most significant explanatory variables were selected by a random partition of the available dataset in two parts (training and test subsets), for 100 times according to a bootstrap procedure. These variables (selected 80 times at least by the bootstrap procedure) were used to build the final susceptibility model, whose accuracy was estimated through a 100-fold repetition of holdout method for regression based on the training and test sets created through the 100 bootstrap model selection. The presented methodology allows the identification, in a robust and reliable way, of the most susceptible road sectors that could be hit by sediments delivered by landslides. The best predictive capability was obtained using a model in which also the index of connectivity, calculated according to a linear relationship, was considered.Most susceptible road traits resulted to be located below steep slopes with a limited height (lower than 50 m), where sediment connectivity is high. Different land use scenarioswere considered in order to estimate possible changes in road susceptibility. Land use classes of the study area were characterized by similar connectivity features. As a consequence, variations on the susceptibility of the road network according to different scenarios of distribution of land cover were limited. The results of this research demonstrate the ability

of the developed methodology in the assessment of susceptible roads. This could give to the managers of an infrastructure information on the criticality of the different road traits, thereby allowing attention and economic budgets to be shifted towards the most critical assets, where structural and non-structural mitigation measures could be implemented.

**Keywords:** roads, shallow landslides, susceptibility

**1 Introduction**

Landslides are important geohazards in many regions of the world. They cause severe economic damages each year in the order of hundreds of billions dollars (Zezere et al., 2007; Salvati et al., 2014; Gariano and Guzzetti, 2016). Slope instability induces significant damages, deaths and economic losses to infrastructures, in particular to roads (Van Westen et al., 2006; Klose et al., 2015). The main negative consequences of instability phenomena on roads are (Bil et al., 2014): i) their partial

or complete destruction, which can also provoke human losses; ii) the traffic restriction due to the blockage of a hit road, which may affect the entire network causing congestion; iii) the cut-off of certain areas that cannot be reached by alternative routes.

Thus, it is fundamental to identify what sectors of a road network are more susceptible to landslides, in order to reduce the risk to the population potentially exposed and the money expense caused by road damaging. This aim is particularly

important, also because several researches (Nemry and Demirel, 2012; Michaelides, 2014; Strauch et al., 2015; Klose et al., 2017; Matulla et al., 2017) stressed that the exposure of road networks to slope instabilities could increase as a consequence of the climate change and of the economic rising income in different countries.

According to the geomorphological and triggering features, landslides affecting roads can be distinguished in: i) landslides in correspondence of the infrastructure; ii) landslides triggered in a natural or an engineered hillslope upstream to the road,

whose transportation and/or accumulation zone hit the infrastructure.

The triggering mechanisms of the first landslide type are strictly related to local hydrological and geotechnical settings that are related to the road presence. These factors generally highlight an incorrect construction or management of the infrastructure regardless of the natural features of the slopes where the road was built (Sidle and Ochiai, 2006; Muenchow et al., 2012; D'Amato Avanzi et al., 2013; Brenning et al., 2015). On the other hand, the triggering mechanism and landslide

runout of the second landslide type can be related to the geological, geomorphological and hydrological predisposing factors of the natural or engineered slopes upstream to the roads. Furthermore, these events are the most widespread in terms of affected routes, causing in many cases the involvement of extended sectors of hilly and mountainous road networks (Quinn et al., 2010; Bil et al., 2014).

In recent years, several data-driven methodologies were built to identify the susceptible sectors of a road network towards

landslides (Budetta, 2004; Hearn et al., 2008; Jaiswal et al., 2010a, 2010b, 2011; Quinn et al., 2010; Michoud et al., 2012; Tarolli et al., 2013; Bil et al., 2014, 2017; Penna et al., 2014; Ramesh and Anbazhagan, 2015; Tarolli and Dalla Sofia, 2016; Winter et al., 2016; Donnini et al., 2017; Pellicani et al., 2017; Postance et al., 2017; Martinovic et al., 2018). These methods

are based on quantitative statistical relationships between predisposing factors and a response variable. They assume that an event is most likely to occur under similar ground conditions to previous events (Varnes, 1984). They present the advantage of being more objective and easily applicable at different scales (from site-specific to regional), as well as capable of managing large sets of predisposing factors (Corominas et al., 2014).Data-driven models depend strictly on the reliability of the inventories of the response variable (Guzzetti et al., 2006; Corominas et al., 2014). Besides this limitation, data-driven are most flexible to be used at different scales of analysis (from site-specific to regional scale) and do not require a lot of data not easily to be estimated as for the physically-based models (Corominas et al., 2014).

Data-driven models used for the characterization of susceptible routes were based on a multivariate analysis (Dai and Lee 2002; Chen and Wang 2007), which predicts the spatial distribution of roads hit by landslides through the estimation of the relations and the relative weight between the predisposing factors and the response variable (roads affected by landslides). Such methods do not consider non-linear relations between the predisposing factors and the response variable. However, the non-linearity of the system should be considered, since the changing in the environmental and geological conditions leads to a consequent interaction of the mobilized materials with roads (Goetz et al., 2011). Moreover, neglecting a possible non-linearity in the model could decrease its predictive performance, due to a limitation in highlighting the complex behaviors of the phenomena (Phillips 2003, 2006). Thus, it could be useful implementing a methodology that considers also a non-linear regression technique, such as the Generalized Additive Model (GAM; Hastie and Tibshirani, 1990).

Furthermore, previous methods did not take into account for the potential slope sediments, mobilized by the landslide, to reach the road network in downstream area. This aspect is well described by the amount in sediment connectivity, which influences the path and the travel distance of the materials mobilized by a slope failure till a potential natural or anthropogenic barrier (e.g. river or road) (Cavalli et al., 2013; Tarolli and Sofia, 2016; Persichillo et al., 2018). In this way, the landslide runout can be estimated and inserted in modeling roads susceptibility, without employing numerical or physically-based methods which require several rheological and geotechnical data not easily measurable for the slope materials (Hungr, 1995; Fannin and Wise, 2001; Pastor et al., 2014; Fan et al., 2017).

Scenarios of road susceptibility distributionrelated to the modifications of land use in a particular area were not considered so far. However, land use changes can have significant impacts both on the locations of landslides triggering zones (Glade, 2003; Begueria, 2006; Reichenbach et al., 2014; Persichillo et al., 2017a) and on the connectivity of the mobilized sediments (Foerster et al., 2014; Lopez-Vicente et al., 2013, 2016). Thus, susceptibility scenarios of different land use distributions may allow to identify land management practices able to reduce the slope instability which can induce damages to roads.

For these reasons, a non-linear data-driven method was developed and tested for the identification of road network sectors more susceptible to shallow landslides triggered in slopes upstream to an infrastructure. The main objectives of the paper are: i) the development and the test of a data-driven non-linear methodology, based on the GAM, able to identify the relations between predisposing and response variables for the assessment of the road sectors susceptible to shallow landslides triggered in slope upstream to the infrastructure; ii) the importance of considering sediment connectivity in the

susceptible road segments modelling; iii) the analysis of the effects resulting from different scenarios of land use distribution on the road sectors potentially affected by shallow landslides.

## 2 The study area

The analysis was carried out in a catchment located between Scuropasso river and Versa river, in OltrepòPavese, in the northern termination part of the Italian Apennines (Fig. 1). The study area is 14 km$^2$ wide and presents an elevation range between 88 and 295 m a.s.l. The morphological structure is typical of the Pede-Apennine margin of OltrepòPavese and is closely related to both the lithology and the tectonic/neotectonic setting of the Apennine margin. It is characterized by a medium-high slope gradient, with slope angles higher than 10°, with prominent altimetric irregularities along ridge lines and channel network in narrow valleys (Bordoni et al., 2015). Bedrock is characterized by a Mio-Pliocenic succession formed by medium low-permeable arenaceous conglomeratic materials (Monte Arzolo Sandstones, Rocca Ticozzi Conglomerates) overlying impermeable silty-sandy marly bedrock (Montù Beccaria Formation, Sant'AgataFossili Marls) and evaporitic chalky marls and gypsum (Gessoso-Solfifera Formation) (Vercesi and Scagni, 1984). Superficial soils, derived by bedrock weathering, are mainly clayey-sandy silts and clayey-silty sands. Soil depth has values lower than 2.5 m.

Land use maps of the study area have been available since 1954. Land use map of 1954 was realized by aerial photographs from GruppoAereoItaliano (Italian Aerial Group), with a resolution of 0.5 m. Moreover, the land use map of 1980 was obtained from photo interpretation at a scale of 1:50,000 from the TEM1 flight (scale 1:20.000). Land use maps of 2000, 2007, 2012 and 2015 were provided by the Lombardy Region and shared as part of the Infrastructure for Spatial Information in Lombardy (IIT) via the Geoportal (Lombardy Region Geoportal: http://www.cartografia.regione.lombardia.it/geoportale, last access: 11 December 2017). The map of 2000 was obtained from the photo interpretation of aerial images of Flight IT2000, with a resolution of 1 m. While, the land use map of 2007 was realized by using colour and infrared orthophotos from Flight IT2007, with a resolution of 0.5 m. The maps of 2012 and 2015, which corresponded to the actual situation, were realized through the photo-interpretation of aerial photos realized by Agency for Disbursement in Agriculture (AGEA). The photo-interpretation was also supported by auxiliary data of Lombardy Region databases (e.g. Regional Agricultural Information System, Forest Types maps, map of the resident population, Archive of Integrated Activities production). The overall accuracies of maps obtained for Lombardy Region using this methodology was reported in Zaffaroni (2010) as approximately 95%. More detailed information about the method to realize these maps are available in Fasolini (2014).

The study area is characterized by traditional viticulture vocation, with grapevine cultivation that represents the main economic branch. Till the 1980s, more than 90% of the territory was covered by cultivated vineyards, where manual cultivation practices predominated (Fig. 2a, b, c). This situation represented the highest diffusion of vineyards in the study area, identifying all the hillslopes that are effectively adapt for grapevine implantation and cultivation. Instead, in the last 40 years, more than 40% of previously cultivated slopes were abandoned, with a correspondent progressive increase in woodlands (+13% from 1980 to 2007–2015) and in uncultivated areas generally composed by shrubs and grasses (+10%

from 1980 to 2007–2015) (Fig. 2a, d). Between 2007–2015, land use classes distribution kept steady. In 2007–2015 time span, 49% of the area was occupied by vineyards, 10% by uncultivated areas, 16% by woodlands and 16% by urban areas (Fig. 2d). Other land use classes are present in a percentage lower than 5%.

This abandonment was due to the conversion from manual to mechanical cultivation practices, that increased the difficulties in the maintenance of vineyards, especially for those located on very steep slopes (> 25°) (Persichillo et al., 2017a). Moreover, societal changes, together with the decreasing number of people actively cultivating the area, caused a reduction in land care practices and maintenance works in both abandoned and still cultivated vineyards (Persichillo et al., 2017a, 2018).

A primary road network (81 km long and generally 3–5 m wide) crosses the study area and is composed of provincial and municipal roads that connect different villages and towns (Fig. 3). The roads were built in correspondence of the valley floors or in the medium part of a hillslope, cutting its continuity. In the second case, a 3–5 m height trench was built upstream to the road sector.

This area was recently affected by several shallow landslides triggered by intense rainfall events (Bordoni et al., 2015). The most important one occurred in 27–28 April 2009 (160 mm/62 h), and induced 532 failures. Other shallow landslides occurred during the events of March/April 2013 and of 28 February–2 March 2014, triggering 19 and 18 shallow failures, respectively. Lower numbers of phenomena reflect the lower amount of rainfall recorded during these events (40 mm in about 30–-50 h in March/April 2013 events; 69 mm in 42 h in 28 February–2 March 2014 event).

These landslides had an average length of about 35 m and area varied from a minimum of 13 $m^2$ to a maximum of almost 9,000 $m^2$, with an average of about 477 $m^2$. The failure surface was mainly detected between 0.9 and 1 m from the ground level, generally in correspondence between the soil-bedrock contact. 30% of these shallow landslides were triggered in vineyards, while an equal percentage of phenomena developed in woodlands or uncultivated areas. According to Cruden and Varnes (1996)' classification, most of the shallow landslides can be classified as roto-translational slides evolved into flows, with width/length ratio > 1. Moreover, 24 failures (5% of the total number), were roto-translational slides affecting the trench in correspondence of a cut of a road.These phenomena were named as B2 type, according to the term used by Zizioli et al. (2013) and Persichillo et al. (2018).

The landslides significantly affected the road network with severe damage (Fig. 3) regarding the partial or complete destruction of road traits, debris accumulation and blockage, causing traffic restriction and the cut-off of villages and towns.

A detailed inventory map of the road sectors affected by shallow landslides in the study area was prepared and used as response variable of the model. The inventory map of the affected road traits include all the sectors hit by the shallow landslides occurred in the study area during 27–28 April 2009, March/April 2013 and 28 February–2 March 2014 rainfall events. For 2009 event, color aerial photographs at a resolution of 15 cm acquired immediately after the event were examined (Persichillo et al., 2017a). For 2013 event, affected road traits were identified by visual interpretation of Pleiades satellite images with a resolution of less than 1 m (Persichillo et al., 2017a). For 2014 event, slope failures and affected roads

immediately after the event were detected through field surveys; the identified phenomena were mapped through a GPS tool, whose resolution is less than 2.5 m.

In particular, 2.5 km of the principal road network was affected by shallow landslides in the last years. 134 shallow landslides (23%) hit roads.24 failures (15% of the total number) were roto-translational slides affecting the trench of a cut realized for building a road. Instead,the remaining 90 phenomena (85%) were shallow landslides triggered in slopes upstream the routes on cultivated or abandoned hillslopes. The length of the road sectors hit by a shallow landslide ranged between 2 and 94 m.

## 3 Methods

### 3.1Development and test of the data-driven model

#### 3.1.1Predictor variables

A data-driven methodology based on GAM was implemented for the assessment of roads that could be hit by shallow landslides. A schematic flow-chart of this methodology is shown in Fig. 4.Such procedure is similar to that one proposed by Persichillo et al. (2017b) for the assessment of the shallow landslide susceptibility in different settings. In this paper, it was refined for the application to roads susceptibility towards landslides. In particular, different predictor variables and response variables were considered, according to their influence on the possible interaction between landslide mobilized materials and the road network located downstream.

In the model, 11 predictor variables were identified. 8 of these parameters were extracted by a 1 m resolution LiDAR-derived Digital Elevation Model (DEM), through SAGA GIS (System for Automated Geoscientific Analyses; Olaya, 2004; Conrad et al., 2015). The DEM was available from the Italian Ministry of Environment and Protection of the Land and Sea, following the realization of the Piano Straordinario di TelerilevamentoAmbientale (Extraordinary Plan of Environmental Remote Sensing - PST-A). These attributes were: slope angle (SL), aspect (ASP), curvature (CURV), slope length (LEN), slope height (HEI), catchment area (CA), catchment slope (CS) and topographic wetness index (TWI).

SL, ASP, and CURV were calculated through Zevenbergen and Thorne (1987)'approximations. SL strongly controls the velocity of the material mobilized by a shallow landslide, thus its capacity of travelling for long distances from the source areas (Fannin and Wise, 2001; Catani et al., 2013; Fathani et al., 2017). ASP influences the soil moisture and the vegetation growth, that can have a key role on the susceptibility of a slope to shallow failures (Van Westen et al., 2008; Jaiswal et al., 2010a). CURV influences the amount of water runoff, the rate of underground water movement and the potential rates of sedimentation and erosion (Dai et al., 2002; Kritikos and Davis, 2015).

LEN and HEI are key parameters for the estimation of the distance travelled by a landslide from its source area and of the velocity of the displaced material (Bathurst et al., 1997; Chau et al., 2004; Martinovic et al., 2016). LEN is the distance from the point of origin of overland flow to the point where either the slope gradient decreases enough for deposition to start, or runoff waters are streamed into a channel (Wischmeier and Smith (1978). This parameter is useful to predict zones where the

soil deposition is predominant (Winchell et al., 2008). HEI represented the elevation difference between the source area of a shallow landslide and the bottom of the hillslope where this failure occurred.

Multiple-flow direction algorithm (Quinn et al., 1991) was used to obtain CA and CS. Multiflow direction algorithm distributed the water flow to all neighboring downslope cells weighted according to slope angle, avoiding the flow

concentration to particular lines sometimes unrealistic. In the case of planar and concave hillslopes, as the ones present in the study area, the partitioning of the flow provided by the use of the multiflow direction algorithm was consistent to the real situation (Seibert and McGlynn, 2007).

CA is used as a proxy for soil moisture and soil depth, thus for the potential amount of materials that can be mobilized by the shallow landslide and that can reach an infrastructure (Brenning et al., 2015). CS influences the destabilizing forces upstream

that can provoke the development of a landslide (Brenning et al., 2015; Persichillo et al., 2017). TWI highlights the water fluxes along the slopes and the position of the accumulation points in a catchment (Seibert et al., 2007).

Along with the DEM-derived predictor variables, the Euclidean distance from shallow landslide source area (DIST) was calculated, considering the shortest distance between the landslide source area and a considered road trait. The choice of an Euclidean distance was consistent to the types of slope failures present in the study area. The shallow landslides did not

follow established paths of the flow direction on the hillslopes where they occurred. Moreover, they were not channeled, as in the case of typical debris flows. Furthermore, a distance calculated along the flow direction was not considered to avoid redundancy with the parameter of sediment connectivity. In fact, sediment connectivity already took into account for the shortest paths along the flow direction in its downslope component (Cavalli et al., 2013; Crema and Cavalli, 2018). DIST parameter is consistent with slopes of homogeneous gradient, aspect and curvature and is important to understand the

capacity of the mobilized material to travel along a slope and to reach a route located downstream (Bil et al., 2014; Brenning et al., 2015). The source area of each slope failure was extracted through Galve et al. (2015)'s procedure, selecting 25% of the landslide area in correspondence to the highest elevations.

Bedrock geology (GEO) was also considered as predictor. GEO influences the geomechanical, geotechnical, rheological and hydrological properties of the soil, which have effects on the runout of a landslide (Hungr, 1995; Pastor et al., 2014). GEO

was obtained from the geological map of the studied catchment, realized by the Department of Earth and Environmental Sciences of University of Pavia through field surveys.

Different authors (Budetta, 2004; Jaiswal et al., 2010a, 2010b, 2011; Quinn et al., 2010; Michoud et al., 2012; Bil et al., 2014, 2017; Ramesh and Anbazhagan, 2015; Pellicani et al., 2017; Postance et al., 2017) had already used some of the previously described predictorsin different data-driven model aiming to assess roads susceptible to be hit by shallow

landslides. Until now, sediment connectivity has not been considered yet. Tarolli and Sofia (2016) and Persichillo et al. (2018) analyzed two different hilly and mountainous catchments, located in western USA and north-eastern OltrepòPavese, respectively, in order to highlight potential connections between a road network and the sediment delivery. Both the works quantified the sediment connectivity using the index of connectivity (IC), that allows evaluating the potential connection between hillslopes and features which act as targets for transported sediments based only on of the morphological and

topographical characteristics and the vegetation cover of a territory. These works highlighted that the segments of the road network, which can act as a storage area for the sediments mobilized by a phenomenon upstream to the road, are those ones located in correspondence of zones characterized by high IC values. This aspect testifies how slope instability phenomena can actively deliver sediment to particular portions of a road network, producing also damages provoked by the impact of the

mobilized materials with the infrastructure (Sidle et al., 2014; Klose et al., 2015). Persichillo et al. (2018) demonstrated that, in two catchments of OltrepòPavese, the road sectors hit by the materials mobilized by shallow landslides occurred upstream are the ones located close to slopes characterized by the lowest or the highest values of sediments connectivity along the entire catchment. In order to verify the potential influence of this parameter in discriminating the susceptible road sectors, an index of sediment connectivity within the predictor variables of the model was inserted.

The index of sediment connectivity (IC), defined by Borselli et al. (2008), evaluates the potential connection between hillslopes and features which act as targets or storage areas (sinks) for mobilized sediments (e.g., channels, basin outlet, lakes, road network). In the proposed model, IC, calculated according to Cavalli et al. (2013)'s approach, was implemented for a better characterization of surface processes and properties and to exploit a high-resolution DEM. For further details on the changes introduced in the IC calculation following this scheme, we refer to Cavalli et al. (2013) and Crema and Cavalli

15     (2018).

IC is calculated according to Eq. (1) combining the upslope ($D_{up}$) and downslope ($D_{dn}$) components of connectivity, respectively:

$$IC = log_{10}\frac{D_{up}}{D_{dn}} \tag{1}$$

IC can have values in the range of $[-\infty, +\infty]$, with connectivity increasing for larger IC values (Cavalli et al., 2013). IC was

calculated through the stand-alone application SedInConnect 2.3 (Crema and Cavalli, 2018).

In the calculation of IC, both the $D_{up}$ and $D_{dn}$ depend on a weighting factor (W) (Eq. 2, 3):

$$D_{up} = \overline{WS}\sqrt{A} \tag{2}$$

$$D_{dn} = \sum_i \frac{d_i}{W_i S_i} \tag{3}$$

where, S is the average slope gradient of the upslope contributing area, A is the upslope contributing area, di and Si are the

length of the flow path and the slope gradient for the $i_{th}$ cell, respectively.

W that is intended to model the impedance to sediment fluxes was extracted in two different ways:

1) according to the linear formulation of W (Eq. 4) as a function of land use:

$$W_{lin} = 1 - n \tag{4}$$

, where n Overland Flow Manning's n Roughness Value, which depends on the land use type (Tab. 1);

2) according to the non-linear approach proposed by Gay et al. (2015) and Kalantari et al. (2017) as a function of the morphological properties and of the land use characteristics (Eq. 5):

$$W_{nl} = \frac{1}{1+e^{-0.5(x-x_0)}}\left(1 - \frac{R_i}{R_{imax}}\right) \qquad (5)$$

, where RI is the roughness index dependent on the surface morphology variability (Cavalli et al., 2008; Cavalli and Marchi, 2008), $RI_{max}$ is the highest value of RI in the study area, $x_0$ is the midpoint of the distribution function of RI in an area. According to the different ways of calculation, IC distribution changes (Kalantari et al., 2017). In the considered case study, IC was calculated with both the approaches, producing two IC maps ($IC_{lin}$ obtained implementing $W_{lin}$, $IC_{nl}$ obtained implementing $W_{nl}$), inserted alternatively in the model for the assessment of the roads susceptible to shallow landslides. For each trait of the road network analyzed, the value of each assigned predictor corresponded to the one of the slope immediately upstream the road trait, where a landslide, that could hit this sector, could be triggered. This is consistent with the features of the slopes where shallow landslides occurred in past in the study area. In fact, from the source area to the accumulation zone of each landslide, the failed slopes kept similar morphological and hydrological features, in terms of slope angle, exposition, curvature and hydrological features (Bordoni et al., 2015; Persichillo et al., 2017b). Maps of the predictor variables were produced for the study area at a resolution of 1 m, as the input DEM.

### 3.1.2 Response variable

The response variable corresponded to the inventory of the road traits hit by sediments mobilized by shallow landslides during the events occurred between 2009 and 2014.In the inventory map, a binary information was inserted. A value equal to 0 was assigned to the road segments not affected by a shallow landslide, while a value of 1 was assigned to each hit road trait. The resolution of this map was set as the ones of the predictor variable (1 m). The inventory maps referred to the primary road network of the study area, composed of provincial and municipal routes. This was considered because this network contains the most affected road sectors, in terms of economic damages and indirect losses (restriction of traffic, cut-off of villages for the blockage of the road).

### 3.1.3 Implementation of GAM model

The data-driven methodology developed for assessing susceptible roads was based on GAM. GAM is an extension of the Generalized Linear Model (GLM), in which the linear function is replaced by an empirically fitted smooth function that allows fitting the data in the more likely functional form (Hastie and Tibshirani, 1990; Goetz et al., 2011). GAM uses a link function to relate the mean ($\mu$) of the response variables (probability that a road sector could be hit by a landslide) and the sum of smooth functions of the predictor variables (Jia et al., 2008) (Eq. 6):

$$g(\mu) = \sum_{i=1}^{n} f_i(x_i) \qquad (6)$$

, where g is the link function and the $f_i$ are smooth function (typically splines), each dependent on a single predictor variable $x_i$ chosen in a set of n variables $x_i...x_n$.

GAM was implemented through 'gam' package of R software (Hastie, 2013).Starting from null model, each predictor variable can be included in the GAM model as linear (untransformed), non-linear (non-parametrically transformed with two equivalent degrees of freedom), or not included in the model. For the selection of the explanatory variables, we used the 'step.gam' command of the R package 'gam'. The variables were selected allowing both directions in the step-wise search, using the option direction="both" in issuing the step.gam command. The selected "best" model is the one that minimizes the Akaike Iteration Criterion (AIC) statistic.

The adopted procedure was composed of the following steps.

The first step was the application of a multicollinear analysis between the numerical predictor variables. Multicollinearity verifies when some predictor variables are linearly correlated among them to avoid redundancy that could affect the numerical stability (Farrar and Glauber 1967). The condition indexes of the matrix of the independent variables was calculated. Variables featuring such an index higher than 30 were considered not independent, thus they were excluded from the analyses to reduce collinearity (Belsley et al., 1980).

In the second step, a database formed of an equal number of road pixels affected or not by shallow landslides was implemented in order to avoid theover-estimation of non-landslide areas, which are much widerthan landslide ones (Dai and Lee 2002; Ayalew and Yamagishi, 2005; Persichillo et al., 2017b). Then, this databasewas subdivided into training and test sets.The training set, corresponding to 2/3 of the dataset, was used to fitthe model.Whereas the test set, forming of the remnant 1/3 of the dataset, was used to verify the accuracy of the model.Training and test sets were randomly selected for 100 times according to a bootstrap procedure. The most frequent predictor variables (selected 80 times at least by the bootstrap procedure) were used to build the final susceptibility model. Moreover, linear and non-linear predictors were identified according to the higher percentage of selection of each parameter.

Model forecasting capability constituted the third step of model scheme. A 100-fold repetition of holdout method for regression with a binary response(McLachlan, 1992; Molinaro et al., 2005; Maindonald and Braun, 2010), consisting of a random sub-sampling of different training and test sets, in the proportion of 2/3 for testing and 1/3 for test, was implemented. The accuracy calculated for these iterations in all training and test sets was averaged to obtain its overall value. The considered training and test sets were the ones created through the 100 bootstrap model selection. The area under the Receiver Operating Characteristic (ROC) curve (AUC) (Hosmer and Lemeshow, 2000) was computed to evaluate the model ability to discriminate affected road sectors, furnishing a further measure of the accuracy of the model. The AUC can take values from 0.5 (no discrimination) to 1.0 (perfect discrimination; Spitalnic, 2004) Moreover, the mean value and the bootstrap 95% confidence intervals of the 100 AUC obtained from the 100-fold bootstrap procedure for the overall accuracy of the model were calculated.

Furthermore, the 100 fitted bootstrap models were used to extend the prediction to the whole road sectors to obtain the distribution of probability. Thus, the map of the susceptibility to be hit by shallow landslides was obtained from the mean

values of each bootstrap distribution of 100 probability values. Also a prediction uncertainty was associated with to each estimated probability was estimated through the calculation of by calculating the bootstrap 95% confidence intervals of the susceptibility. Different classes of probability susceptibility were created, subdividing into 4 intervals the probability values in the susceptibility map: low ($0 < p \leq 0.25$), medium–low ($0.25 < p \leq 0.50$), medium–high ($0.50 < p \leq 0.75$), high ($0.75 < p \leq 1$).

The number of true positives (TP), true negatives (TN), false positives (FP), and false negatives (FN) was further obtained comparing the susceptibility map with the response variable map used to build the model (Jollifee and Stephenson, 2003). For making this comparison, susceptibility values were classified as a binary variable: 1 was assigned to values higher than 0.5 (modeled pixel hit by a landslide), while 0 was assigned to values lower than 0.5 (modeled pixel not affected by a landslide).

Susceptibility was calculated considering a spatial resolution of 1 m, as the input predictors, and for a buffer of 5 m from the middle of each road sector. The chosen buffer of 5 m was consistent with the size of the roads present in the study area. These roads had similar sizes, with a width of the roadway ranging between 3.5 and 5 m.

To assess the effect of considering IC in modeling the susceptibility, three models were produced and compared: Model 1) using all the predictor variables except for the IC; Model 2) considering all the predictors with $IC_{lin}$; Model 3) considering all

the predictors with $IC_{nl}$.

## 3.2 Change in susceptibility according to different land use scenarios

IC depends on the morphological features and on the land use of hillslopes, due to the presence of W factor. On the hypothesis that morphological features does not change, IC maps were created using particular land use distribution, representative of potential situations which could characterize the study area.

Besides the current scenario used for building the susceptibility models at this time, other three scenarios were considered. The second scenario (Scenario 2) consists of the 1980 land use map, where the widest extension of cultivated vineyards was reached (Fig. 2c). Thus, this scenario represents the possible distribution of vineyards in the case of a complete recovery of the abandoned areas since 1980s.

The third scenario (Scenario 3) corresponded to the actual scenario, with an interruption in the increase of abandoned areas

without the recovery of the previously cultivated slopes. According to this, uncultivated areas completely disappear and they convert into woodlands (Fig. 5a). This scenario is consistent with the new land use management policies that were developed at the municipal level in the study area, aiming at regulating the diffusion of uncultivated areas (Rural Police Regulation, 2008; Persichillo et al., 2017a).

The fourth scenario (Scenario 4) corresponded to a further increase in the abandonment of cultivated grapevines (Fig. 5b).

According to this, actual uncultivated areas transform into woodlands, while further uncultivated ones develop in correspondence of actual vineyards. The slopes where abandonment was supposed are the currently cultivated ones with similar morphological features (slope angle higher than 15°) to the abandoned areas in the period 1980–2015. The increase in abandoned areas was kept equal to 22%, as that one occurred from the period 1980–2015.

Different IC scenarios were then created using these land use distributions and they were inserted in GAM model for assessing the susceptibility change of road traits in function of this parameter. Other morphological and hydrological input predictors were kept steady. The model used for these reconstructions corresponded to the one that had the best predictive performance considering the actual situation.

## 4 Results

### 4.1 Map of IC reconstructed through linear and non-linear methodology

The distribution of IC for the actual conditions, reconstructed through the linear ($IC_{lin}$) and non-linear ($IC_{nl}$) calculation of the W factor, was analyzed (Fig. 6). In the study area, $IC_{lin}$ ranged between –7.00 and 1.75, while $IC_{nl}$ values ranged between –4.20 and 2.23 (Fig. 7). The average value of IC distribution was –3.17 for $IC_{lin}$ and –3.57 for $IC_{nl}$, while the standard deviation was similar for both the distributions (0.72 and 0.65, respectively). The map obtained with the linear implementation of W in IC calculation showed values averagely higher than the ones obtained with the non-linear W methodology in the corresponding sectors (Fig. 6).

In these analyses, IC values were classified into four classes (low, medium-low, medium-high, high), by identifying classes limits that best grouped similar values and maximized the differences between classes using the Jenks' natural breaks (Jenks, 1967), following the approach used in similar contexts by Surian et al. (2016), Tarolli and Sofia (2016) and Tiranti et al. (2016).

$IC_{lin}$ map highlighted that the northern and western parts of the catchment were characterized by medium-high and high connectivity (Fig. 6a). All the slopes with a high gradient (generally higher than 15°) presented medium-high and high connectivity features. Highest values were reached in road trenches with limited slope height (lower than 20 m) and at the bottom of hillslopes characterized by high slope angle (higher than 15–20°) and by slope height in the order of 35–70 m. Instead, $IC_{nl}$ map indicated lower connectivity in all the sectors of the study area (Fig. 6b). Where $IC_{lin}$ map showed a wide diffusion of slopes with medium-high and high connectivity, $IC_{nl}$ highlighted especially medium-low and low sediment connectivity (Fig. 6b). Only few areas close to road segments were characterized by high connectivity (Fig. 6d). These sectors corresponded to the road trenches characterized by a slope height lower than 20 m. Both reconstructions showed low and medium-low connectivity where plain areas or hillslopes with slope angle lower than 10° are present (Fig. 6).

### 4.2 GAM models implementation

#### 4.2.1 Selection of the explanatory variables

Three GAM models were tested on the basis of the different set-up of the input predictors. The first phase was the selection of the variables to introduce in each model. All the predictors were not collinear so all these were inserted in the modelling. For each model, the variables whose selection frequency was higher than 80% in the 100-fold bootstrap procedure were selected. The selected variables were the same ones for all three models, with similar selection frequency (Tab. 2). IC was

taken into account only in Model 2 and Model 3, and then consequentially selected in both these models (Tab. 2). Besides IC (having a selection frequency equal to 100% in both Models 2 and 3), the variables selected are the following (Tab. 2): SL (97%), CURV (87%), HEI (88–92%), CS (100%), TWI (85–95%), DIST (100%), GEO (100%) in all the three models. ASP, LEN, and CA were excluded from all the models. Among these variables, only CA had a quite high frequency of selection

(56–66%), but it fell under the defined threshold (Tab. 2). The selected continuous explanatory variables (all the predictors, except GEO) were distinguished into linear or non-linear, on the basis of the higher percentage of selection obtained in the bootstrap procedure. SL and HEI were chosen as linear, while CURV, CS, TWI, DIST, and IC were selected as non-linear (Tab. 2). Despite the different types of calculation of the IC implemented in Model 2 ($IC_{lin}$) and in Model 3 ($IC_{nl}$), this variable was evaluated as significant in both this model, with a frequency of 100%. Moreover, IC was chosen as a non-linear

variable in both these models (Tab. 2).

### 4.2.2 Predictive performance and susceptibility maps of the models

Model 1, that did not consider IC, is characterized by a fair predictive capability. In fact, AUC of the training and the test sets of this model were equal to 0.71 and 0.70, respectively (Tab. 3). AUC of the final susceptibility map produced with Model 1 was similar to those of training and test sets (0.74; Tab. 3). However, the predictive capability increased whether IC

parameter was added among the predictor variables. In particular, AUC of training and test sets increased till 0.82 for Model 3, that considered also $IC_{nl}$. For this model, AUC of the final susceptibility map was of 0.83, with an increase of 0.09 respect to Model 1 (Tab. 3). A better effectiveness was reached if $IC_{lin}$ was taken into account (Model 2). AUC of training and test sets of Model 2 reached values of 0.90, while AUC of the final susceptibility map was of 0.94. According to Spitalnic (2004), a model with similar predictive performances can be classified as excellent.

Table 3 also highlighted very little values of standard deviation of AUC of training and test sets of each model, that maintained equal to 0.01. This confirmed the reliability of the procedure used to build up the different models.

Furthermore, the bootstrap 95% confidence intervals of AUCs were every of 0.02. This result is also confirmed by the very narrow bootstrap 95% confidence bands of ROC curves (Fig. 8a, b, c). The maps showing the bootstrap 95% confidence intervals of the probability for each road trait to be hit by a shallow landslide are illustrated in Fig. 8. As confirmed by the

low values of the confidence intervals, remaining lower than 0.25, the spatial variability of this probability is generally low in the entire road network of the study area for each model.

The predictive capability of the models were also evaluated by computing the values of the four indexes of a four-fold plot. TP and TN were significantly higher in Model 2 than Model 1 and 3, while FP and FN were significantly lower in the same model that the others (Fig. 8d). TP and TN reached values of 90.0 and 84.8% in Model 2, respectively. These values

highlighted an increase of 5.1–13.7% respect to Model 3 and of 13.3–30.6% respect to Model 1. The highest effectiveness of Model 2 was confirmed also by the lowest values of FP and FN (10.0 and 15.2%, respectively), that were lower of 5.4–35.8% than Model 1 and Model 3 (Fig. 8d).

The susceptibility maps for the road network extracted by GAM models are in Fig. 10.Model 1 classified 46.9% of the road network in medium-high and high susceptibility classes. This percentage is significantly higher than those obtained for Model 2 and Model 3 (Fig. 11). The widespread diffusion of high susceptibility areas in Model 1 explains also the high values of FP measured for this model.

Model 2 and Model 3, also considering IC within predictors, classified a lower percentage of the road network in medium-high and high susceptibility classes which are overall of 15.4% and 18.3%, respectively. (Fig. 11). The number of high susceptible road traits of Model 3 seems overestimated respect to the real situations, as demonstrated by the higher FP and FN than Model 2 (Fig. 8d). Model 2 presented a higher predictive performance than the other models, as confirmed by the quantitative indexes calculated for GAM model. It classified 15.4% of the road network of the study area in medium-high and high susceptibility classes and the remnant 85.6% in low and medium-low classes (Fig. 11). All the susceptibility maps classified as more susceptible the road sectors located below slopes of SL higher than 20°, HEI lower than 50 m, catchment slope between 28 and 30° and DIST in the range of 40–100 m. Moreover, Model 2 and Model 3 discriminated as more susceptible those road traits in correspondence of areas with medium-high and high IC, generally higher than –3, regardless of the land use which covered the slope above the road.

## 4.3 Susceptibility maps according to different land usescenarios

The assessment of the predictive capabilities of the GAM models related to the actual scenarios revealed that Model 2 was the best one. This model took into account for several morphological and hydrological features of the slopes upstream the road sectors and the sediment connectivity, evaluated according to the linear modeling of IC parameter. Due to the importance of considering IC distribution in the evaluation of the routes that could be affected by shallow landslides, susceptibility scenarios were created varying IC maps input according to three defined scenarios of land use distribution hypothesized for the study area (Scenario 2, Fig. 2c; Scenario 3, Fig. 5a; Scenario 4, Fig. 5b). In fact, IC may change as a function of the change in the distribution of W factor used in the calculation of this index.

Fig. 12 illustrates the influence of different land use scenarios on IC. Its spatial distribution did not seem to be affected by land use changes presented in the considered scenarios. The connectivity of a particular hillslope kept approximately equal to the actual scenario. This was also confirmed by the mean and the standard deviation of the distribution of IC values in the study area, which remained equal to –3.20/–3.17 and 0.72/0.74, respectively.

The similar maps of $IC_{lin}$ obtained for the different land use maps implicated that the susceptibility distribution along the road network did not change significantly for the different considered scenarios (Fig. 13). Compared to the susceptibility of Model 2 created considering the actual scenario (Fig. 11), the differences on the percentages of the road network classified with low, medium-low, medium-high and high susceptibility by the other reconstructed scenario were negligible, ranging in the order of 0.1–0.2% (Fig. 13).

**5 Discussions**

In this work, a methodology able to classify in different susceptibility classes the traits of a road network potentially hit by sediments of landslides triggered above the road was developed and tested. Different Authors (Budetta, 2004; Hearn et al., 2008; Jaiswal et al., 2010a, 2010b, 2011; Quinn et al., 2010; Michoud et al., 2012; Tarolli et al., 2013; Bil et al., 2014, 2017;

Penna et al., 2014; Ramesh and Anbazhagan, 2015; Tarolli and Dalla Sofia, 2016; Winter et al., 2016; Donnini et al., 2017; Pellicani et al., 2017; Postance et al., 2017; Martinovic et al., 2018) developed similar approaches in other geological/geomorphological settings, basing on the implementation of data-driven techniques for the estimation of road susceptibility. For the first time, the proposed methodology allowed to implement a data-driven technique (GAM method) able to take into account also for the non-linear relationships between the predictors and the response variable (road sector

hit by shallow landslides). Moreover, this model considers also a parameter (the index of connectivity) that, if coupled with a landslide inventory, helps to assess the potential slope sediments mobilized by the landslide triggering which can reach the road network in downstream area, inserting also a proxy of landslide runout in the modeling of roads susceptibility.

The models identified some of the input predictors as non-linear variables (in this case, slope curvature, catchment slope, topographic wetness index, distance from shallow landslides source area, index of connectivity), understanding better the

complex relationships which are present in an area between predisposing factors and susceptible roads (Philips, 2006; Goetz et al., 2011). Moreover, before building the model, the procedure developed for the individuation of the most important predictor variables allowed to improve the knowledge about mechanisms which regulate the location of the damaged roads in such an area, avoiding for collinearity and bias that could reduce the reliability of the susceptibility estimation (Farrar and Glauber, 1967; Hosmer and Lemeshow, 1990; Bai et al., 2010). The robustness of the proposed methodology was also

confirmed by the low confidence degree of AUCs measured for the created models (Petschko et al., 2014). The first reconstructed susceptibility model (Model 1) considers the most important predisposing factors in the study area, chosen among those morphological, hydrological and geological parameters taken into account for these analyses in different contexts by other Authors (Budetta, 2004; Jaiswal et al., 2010a, 2010b, 2011; Quinn et al., 2010; Michoud et al., 2012; Bil et al., 2014, 2017; Penna et al., 2014; Ramesh and Anbazhagan, 2015; Pellicani et al., 2017). The reliability of the model is

quite fair, as testified by its AUC value (0.73) and by its high value of FP and TN indexes (22.3 and 45.8%, respectively). According to Model 1, most susceptible road segments are those located downstream to slopes characterized by high slope gradient (> 20°), limited height (< 50 m), high catchment slope (28–30°) and with shallow landslides triggering zones located very close to the road network (40–100 m). These settings are very widespread in the entire study area (Bordoni et al., 2015; Persichillo et al., 2017b, 2018), but these particular features are not enough to discriminate more accurately those

routes where damages provoked by sediments mobilized by shallow landslides are probable.

Starting from this observation, also IC was inserted in the model for the evaluation of the susceptibility of the roads to be affected by shallow landslides. Other two models were created, differing each other for the type of IC used. Model 2 uses $IC_{lin}$ calculated according to the method proposed by Cavalli et al. (2013), where the W factor in the model is evaluated in a

linear way. Model 3 uses $IC_{nl}$, calculated by means also of a W factor evaluated in a non-linear way and in relation also to the both surface roughness and land use properties of a territory (Fryirs et al., 2007; Cavalli et al., 2008; Cavalli and Marchi, 2008; Gay et al., 2016; Kalantari et al., 2017). In these terms, both $IC_{lin}$ and $IC_{nl}$ represent a structural connectivity depending on the morphological and land use attributes of a territory (Borselli et al., 2008; Cavalli et al., 2013; Crema and

Cavalli, 2015).

The models that consider also sediment connectivity have a higher predictive performance than model 1. This is testified by a high AUC values (0.94 for Model 2 and 0.83 for Model 3) and by higher values of TP and TN (till 90.0% and 84.8%, respectively). Moreover, FP and FN of both these models are lower than Model 1 (till 10.0 and 15.2 respectively).

A sensitivity analysis to assess the role of each predictor variable on the accuracy of the GAM models was performed. This

analysis allowed also evaluating the change in predictive accuracy related to adding or removing a set of predictors according to a threshold of selection different than the used 80% or related to adding or removing a particular predictor. It is important to highlight that the results of this sensitivity analysis shown referred to the susceptibility model which had the best predictive accuracy, that is Model 2. Instead, the quantitative changes on the predictive accuracy related to different sets of predictors were similar also considering Model 1 and Model 3.

Table 4 showed the results of this sensitivity analysis. According to the percentages of selection of each variable in the 100-fold bootstrap procedure (Tab. 2), also thresholds of 50% and 90% of selection frequency were considered and compared to the used threshold of 80%. A threshold of selection frequency lower than 50% was not considered significant.

Considering a threshold equal to 50%, also CA (chosen as a linear variable) had to be inserted for modeling the susceptibility. Instead, the mean predictive accuracy of the model, estimated in terms of AUC value, did not change, for both

the training set, the test set and the final model. The difference in the predictive accuracy was lower than 0.01. Instead, concerning a threshold equal to 90%, CURV, HEI and TWI had to be removed. In this case, the mean predictive accuracy of the best model (Model 2) decreased from 0.90 to 0.84 and from 0.94 to 0.88 for training/test sets and for the final models, respectively. Removing a predictor or a set of these from the susceptibility model caused a decrease of the accuracy due to a reduction in explaining the physical relations between the predisposing factors and the resulting effects on the response

variable, in this case represented by the road sectors hit by shallow landslides. These results demonstrated that a threshold of selection of the predictors equal to 80% allowed to obtain the sets of predisposing factors able to estimate in the best reliable and effective way the susceptibility of the road network to be affected by shallow landslides.

Furthermore, a sensitivity analysis of the different predictors considered as predisposing factors for road susceptibility was performed. This analysis consisted in running one of the models (e.g. Model 2) created considering a threshold of selection

frequency equal to 80%, removing each time one of the selected predictors or adding each time one of the other predictors, whose frequency of selection was lower than 80%. In this way, the sensitivity of the model to each predictor could be quantified.

Removing SL or DIST caused a reduction of the predictive accuracy, for both training sets, test sets and final models, of 0.15–0.16. Instead, this reduction was lower than the one quantified if in the model IC was not taken into account (Model 1).

In fact, the absence of IC provoked a decrease in the accuracy of 0.19–0.20. The removal of CS caused a moderate reduction of the accuracy, correspondent to 0.11. While, removing one of the other chosen parameters (CURV, HEI, TWI, GEO) provoked only a slight decrease in the predictive accuracy, in the order of 0.02–0.06. This meant that they explained less than the other selected predictors the susceptibility of a road to shallow landslides. Instead, CURV values close to the roads

were generally slightly negative (lower than –0.05) and the affected sectors were in correspondence of the lowest CURV values (around –0.40). TWI was generally positive in correspondence of road traits, with values higher than 5 close to sectors affected by shallow landslides. Moreover, damaged road traits were mainly located in areas where GEO was composed of medium low-permeable arenaceous conglomeratic materials (Monte Arzolo Sandstones, Rocca Ticozzi Conglomerates) or impermeable silty-sandy marly bedrock (Montù Beccaria Formation, Sant'AgataFossili Marls).

Moreover, adding alternatively to the chosen predictors one of the other predisposing factors (CA, ASP, LEN) did not modify significantly the reliability of the models. The predictive accuracy improved at most 0.01 for both training sets, test sets and final models. ASP close to the roads was very variable, without the identification of peculiar features. While, LEN and CA values close to the road sectors were in a quite narrow range, between 2 and 150 m and around 102 m$^2$, respectively. The particular distributions of these parameters confirmed their not significant roles in the evaluation of the road

susceptibility.

These results confirmed the significant sensitivity of the susceptibility model to IC, especially the one estimated in a linear way. Neglecting IC in these models caused a big decrease in the effectiveness, which affects significantly the susceptibility classification of the road network. Furthermore, SL and DIST also affected significantly the accuracy of the final susceptibility model and had to be considered for obtaining a correct classification of road network. The models were more

slightly sensitive to the other chosen predictors (CURV, HEI, TWI, GEO). Instead, the leakage of only one of those parameters could decrease the final reliability of the road susceptibility.

It is important to note that the standard deviation of accuracy on training and test sets was of 0.01 for all the models, while the range of the 95 % confidence interval of AUC was of 0.02 for all the models.

The susceptibility maps produced through Model 2 and Model 3 identify the road sectors characterized by the highest values

of IC (IC higher than –3) as the most susceptible. These conditions are measured in several routes regardless of the land cover present in the slope upstream the road. Among these models, Model 2, that consider IC calculated through the linear way, performs better than Model 3. Non-linearly reconstructed IC identified less areas with high connectivity than IC$_{lin}$. Thus, the estimated probability to be affected by sediment impacts is reduced in these road traits. IC$_{nl}$ is more representative of the sediment connectivity in lowland environments, where the connectivity is driven also by other factors (such as the

amount of surface water runoff) together with the morphological features of the hillslopes (Fryirs et al., 2007; Gay et al., 2016; Kalantari et al., 2017). Its application in geomorphological settings characterized by a predominant hilly or mountainous morphology, such as the considered catchment, can implicate an underestimation on the connectivity or disconnectivity of the sediments, influencing also the correct assessment of the sectors of an infrastructure threatened by the material mobilized after a triggering event.

The comparison of the best susceptibility model (Model 2) with the distribution of real case of road sectors damaged by sediments mobilized by shallow landslides triggered upstream the road has confirmed an excellent predictive performance of this model. It allows to identify correctly both the road sectors hit by the accumulation zones of roto-translational shallow landslides triggered in the trenches present in halfway roads (B2 type) and the road traits affected by the materials mobilized by shallow landslides triggered in the slopes upstream the routes in correspondence of cultivated or abandoned hillslopes (Fig. 14). This reveals the suppleness of the methodology to estimate in a reliable way the most susceptible sectors of a road network also in the case of sediment source areas, represented by slope instabilities with different features.

Nevertheless, Model 2 classifies wrongly some road sectors in the study area, as testified by the 15.2% of FN and of 10% of FP cases. FN are mostly located in few pixels, sometimes close to other road traits identified with medium-high or very-high susceptibility. These situations could be linked to local peculiar factors, which may affect road susceptibility, that are not completely described by the input predictors chosen for the model. On the contrary, FP cases correspond to road segments where high susceptibility (higher than 0.5) was estimated. These sectors are mostly located near to traits already affected by shallow landslides materials in past events (Fig. 15). They are in a buffer of less than 250 m, in particular between 50 and 200 m, respect to sectors hit in past, and they present morphological and connectivity features similar to threatened traits. Hence, they could represent sectors which could be affected by future events occurred in the same study area, whether the settings of this zone and the triggering conditions will keep similar to past events. In these terms, susceptibility map obtained from Model 2 is useful in determining accurately the susceptible sectors of a road network, furnishing an important tool for the management of the hazard and for sketching policies of risk reduction out.

Due to the importance of sediment connectivity on the model capability, scenarios of susceptibility were reconstructed, through Model 2, starting from different IC maps were obtained considering particular land use distributions. In fact, changes in land use cause are represented by changes in W parameter of the IC calculation, provoking a potential variation in the connectivity distribution. The distribution of susceptibility and of the roads most probably affected by shallow landslides do not change significantly from the actual situation for the three different modeled scenarios (recovery of all cultivated vineyards, break on the abandonment, further increase of the abandoned areas). This is due to the similar values of W (0.6–0.8) characterizing the most widespread land covers of the study area, which thus induce to a limited change in IC value passing from a land use class to another one. Instead, changes in land use distribution could have effects also on the physical morphology of the hillslopes (Fu et al., 2006; Tarolli et al., 2015). For example, the recovery of the cultivation of grapevines in a slope could lead to the development of a drainage system of the superficial and of the shallow waters and to modification on the slope morphology for the implantation of the vineyards. While the abandonment of previously cultivated vineyards induces changes in flow direction and regulation, with direct consequences on sediment production and delivery (Cevasco et al., 2014; Lieskovsky and Kenderessy, 2014; Tarolli et al., 2014; Prosdocimi et al., 2016). These actions could influence the movement of the rainwater and of the sediment mobilized by runoff, then reduce the connectivity and also the potential susceptibility of a road located downstream. Hence, more detailed scenarios of susceptibility changes in relation to

land use changes will take into account also for the morphological modifications linked to these changes, using also a higher resolution DEM (less than 1 m).

## 6 Conclusions

In this work, a non-linear data-driven approach, based on GAM, was developed for the evaluation of the susceptible road sectors of a network that could be affected by the sediments delivered from shallow landslides occurred upstream. The methodology assessed also the role of the sediment connectivity on susceptibility estimation, by the implementation of the index connectivity calculated according to a linear or a non-linear approach.

Besides the use of an inventory of road damages referred only to three triggering events occurred from 2009 to 2014, the random partition of the entire dataset in two parts (training and test subsets), within a 100-fold bootstrap procedure, allowed to select the most significant predisposing variables. This provided a better description of the occurrence and distribution of the road sectors potentially susceptible to damages induced by shallow landslides.

The best predictive capability was reached by a model which took into account also the index of connectivity, calculated according to a linear way. This index well represented the rates of connectivity and disconnectivity in the studied catchment, in relation to its morphology (steep slopes, narrow valleys) and land uses (vineyards, abandoned areas, woodlands). Most susceptible road traits resulted in the ones located below steep slopes with a limited height (lower than 50 m), where sediment connectivity is high, regardless of the land use which covered the slope above the road.

Different scenarios of land use were implemented in order to estimate possible changes in road susceptibility. Land use classes of the study area were characterized by similar effects on connectivity features.The index of connectivity did not change significantly with a consequent leakage of variations also on the susceptibility of the road networks. Larger effects on sediment connectivity could be induced by modifications in the morphology of the slopes (e.g. drainage system, modification of the slope angle) provoked by the abandonment or by the recovery of cultivations. Then, this could have effects on the sediment delivery and also on the susceptibility of a road to be hit by sediments mobilized upstream.

The presented methodology allows to identify the most susceptible road sectors that could be hit by sediments delivered by landslides in a robust and reliable way. This tool can represent a fundamental starting point for improving the land management of the slopes where the source areas of the sediments could develop, in order to reduce the damages to the infrastructure and the related risks and economic losses. Moreover, the results of the susceptibility analysis can give asset managers indispensable information on the relative criticality of the different road sectors, thereby allowing attention and economic budgets to be shifted towards the most critical assets, where structural and non-structural mitigation measures could be implemented.

Furthermore, thanks to the flexibility of the model in the selection of the predictors, the proposed model can be applied to areas with different geological, geomorphological and land use features, identifying the most important predisposing factors peculiar of each catchment. This method can be also implemented in areas characterized by much larger catchments than the

ones analyzed herein, with the only limit of the availability of high-resolution DEMs and of computational resources. Moreover, the methodology can be applied for estimating the susceptibility and the risks related to landslides affecting other main lifelines, such as railways, gas/oil pipelines, power lines.

## Acknowledgements

The authors wish to thank the anonymous reviewers for their suggestions and contributions to the work.

## Authors contribution

Massimiliano Bordoni analyzed the data, developed the methodological approach and prepared the manuscript; Maria Giuseppina Persichillo helped in the development of the methodology and in the interpretation of the results; Claudia Meisina helped in the interpretation of the results and provided guidance and support throughout the research process;

Stefano Crema and Marco Cavalli helped in building the connectivity scenarios and in supporting their interpretation; Carlotta Bartelletti, Yuri Galanti and Michele Barsanti helped in the development of the data-driven approach and in the interpretation of the results; Roberto Giannecchini and Giacomo D'Amato Avanzi collaborated in writing the manuscript and in the comprehension of the analyses.

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

**Table 1: Overland flow Manning's n Roughness Values assigned to each class of land use maps available for the calculation of $W_{lin}$ factor.**

| Land use classes | Manning's n (-) |
|---|---|
| Woodlands | 0.40 |
| Uncultivated areas | 0.35 |
| Grasslands | 0.25 |
| Orchards/Arable areas/Vineyards | 0.20 |
| Bare soil | 0.05 |
| Urban areas | 0.02 |

**Table 2: Frequencies (in %) of explanatory variables (both linear and non-linear) selected by 100-fold bootstrap procedure. The explanatory variables selected by 100-fold bootstrap procedure with an absolute frequency greater or equal than 80% are highlighted in bold red. In the brackets, the frequencies (in %) of selection of each variable as linear or non-linear is shown. The underlined number corresponded to the frequency of the selected function connected to each variable. SL: slope angle; ASP: aspect; CURV: curvature; LEN: slope length; HEI: slope height; CA: Catchment area; CS: Catchment slope; TWI: topographic wetness index; DIST: distance from the source area of a shallow landslide; GEO: bedrock geology; IC: index of connectivity.**

| Model | SL | ASP | CURV | LEN | HEI | CA | CS | TWI | DIST | GEO | IC |
|---|---|---|---|---|---|---|---|---|---|---|---|
| 1 (Lin.- Not Lin.) | **97 (95-2)** | 2 (0-2) | **87 (41-46)** | 36 (36-0) | **88 (45-43)** | 56 (56-0) | **100 (0-100)** | **85 (14-71)** | **100 (0-100)** | **100** | - |
| 2 (Lin.- Not Lin.) | **97 (95-2)** | 18 (12-6) | **87 (41-46)** | 19 (19-0) | **88 (45-43)** | 66 (66-0) | **100 (0-100)** | **85 (16-69)** | **100 (0-100)** | **100** | **100 (5-95)** |
| 3 (Lin.- Not Lin.) | **97 (95-2)** | 18 (11-7) | **87 (41-46)** | 19 (19-0) | **92 (53-39)** | 65 (65-0) | **100 (0-100)** | **95 (12-83)** | **100 (0-100)** | **100** | **100 (2-98)** |

**Table 3: Mean and standard deviation of accuracy for the training sets, the test sets and the final application of the model to the entire study area.**

| Model | Mean accuracy of training sets (-) | Standard deviation of accuracy on training sets(-) | Mean accuracy on test sets (-) | Standard deviation of accuracy on test sets(-) | Mean AUC of the model(-) | 95 % confidence interval of AUC of the model(-) |
|---|---|---|---|---|---|---|
| 1 | 0.71 | 0.01 | 0.70 | 0.01 | 0.74 | 0.73-0.75 |
| 2 | 0.90 | 0.01 | 0.90 | 0.01 | 0.94 | 0.93-0.95 |
| 3 | 0.82 | 0.01 | 0.82 | 0.01 | 0.83 | 0.82-0.84 |

**Table 4: Sensitivity of the different predictor variables on the accuracy for the training sets, the test sets and the final application of the model to the entire study area. The standard deviation of accuracy on training and test sets was of 0.01 for all the models, while the range of the 95 % confidence interval of AUC was of 0.02 for all the models.**

| GAM model | Mean accuracy of training sets (-) | Mean accuracy of test sets (-) | Mean AUC of the model (-) |
|---|---|---|---|
| 1 | 0.71 | 0.70 | 0.74 |
| 2 | 0.90 | 0.90 | 0.94 |
| 3 | 0.82 | 0.82 | 0.83 |
| 2 - (CURV, HEI, TWI) [threshold of selection equal to 90%] | 0.84 | 0.84 | 0.88 |
| 2 + (CA) [threshold of selection equal to 50%] | 0.90 | 0.90 | 0.94 |
| 2 - (SL) | 0.74 | 0.74 | 0.78 |
| 2 - (CURV) | 0.87 | 0.87 | 0.92 |
| 2 - (HEI) | 0.85 | 0.77 | 0.91 |
| 2 - (CS) | 0.79 | 0.79 | 0.83 |
| 2 - (TWI) | 0.88 | 0.88 | 0.92 |
| 2 - (DIST) | 0.75 | 0.75 | 0.78 |
| 2 - (GEO) | 0.84 | 0.85 | 0.88 |
| 2 + (ASP) | 0.90 | 0.90 | 0.94 |
| 2 + (LEN) | 0.90 | 0.91 | 0.95 |

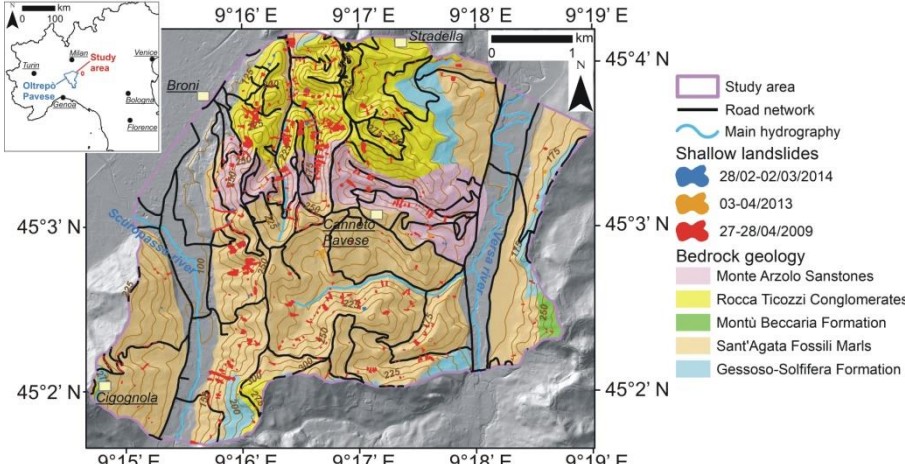

**Figure 1: Geological setting and shallow landslides distribution of the study area.**

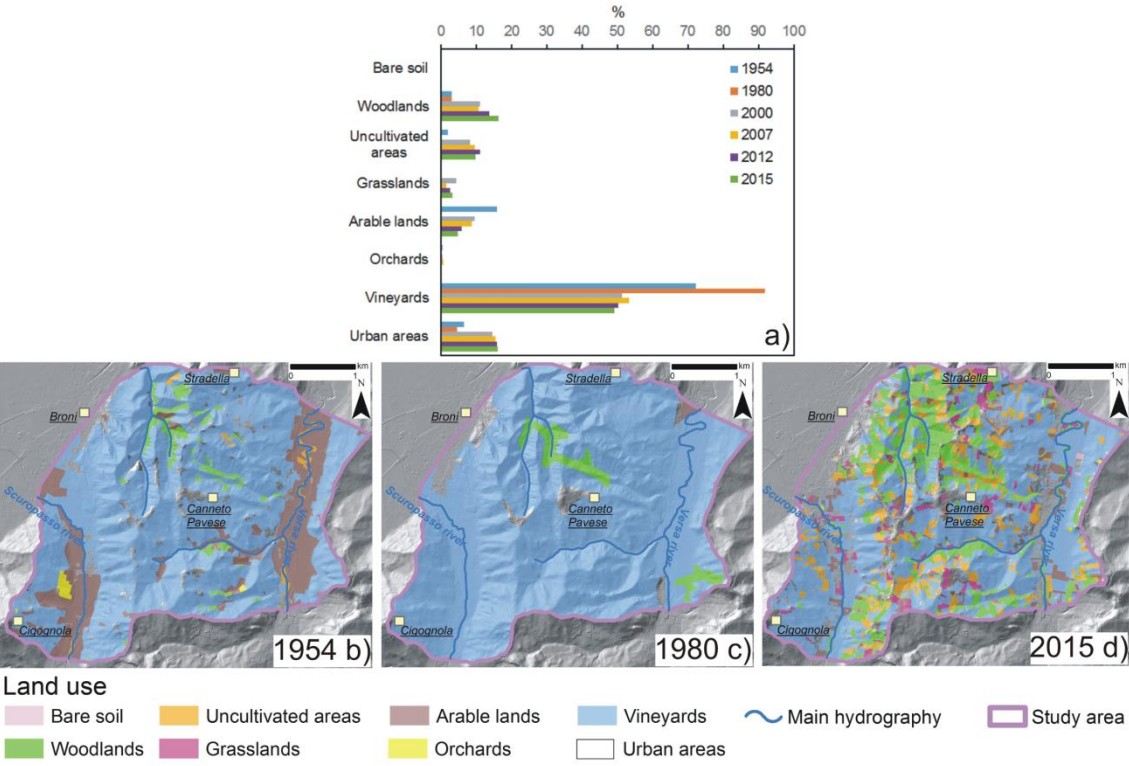

**Figure 2: Land use distribution and land use changes in the period 1954-2015: a) percentage of the area occupied by each land use class during the analyzed period; b) land use distribution in 1954; c) land use distribution in 1980; d) land use distribution in 2015. Land use maps were provided by the Lombardy Region and shared as part of the Infrastructure for Spatial Information in Lombardy (IIT) via the Geoportal (Lombardy Region Geoportal: http://www.cartografia.regione.lombardia.it/geoportale, last access: 11 December 2017). The detailed information regarding the method to realize these maps are available in Fasolini (2014).**

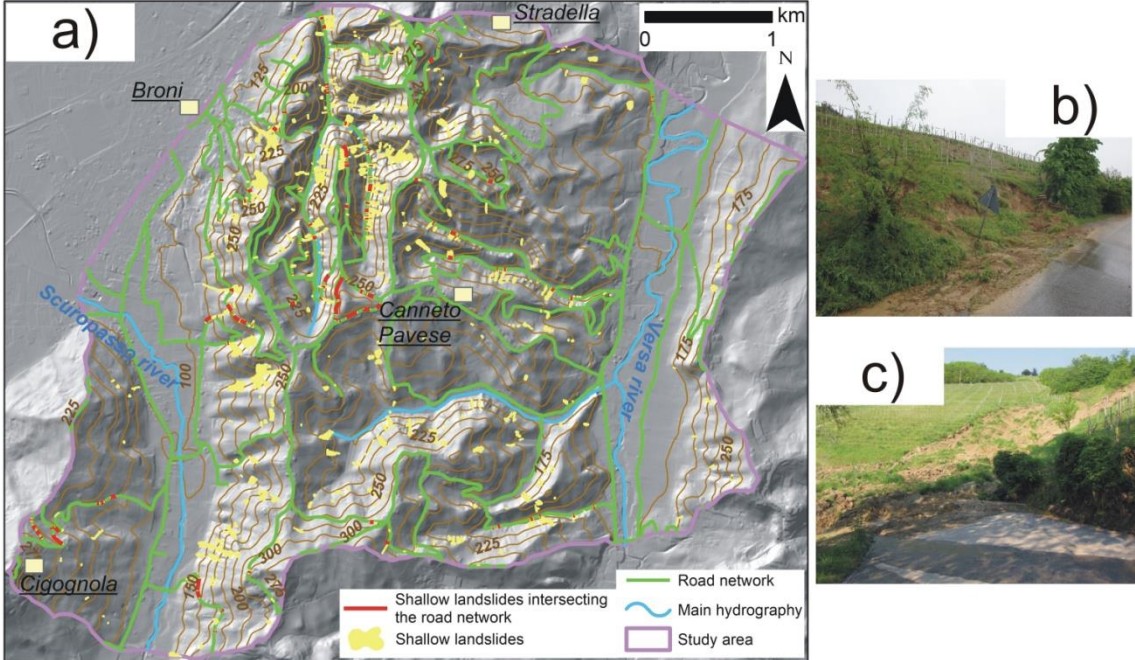

**Figure 3: a) Primary road network of the study area, with the shallow landslides events occurred between 2009 and 2014 which affected these routes. This road network is composed of provincial and municipal roads (available from: Administration of Pavia Province and Infrastructure for Spatial Information in Lombardy (Lombardy Region Geoportal: http://www.cartografia.regione.lombardia.it/geoportale, last access: 11 December 2017). b) A shallow landslide (B2 type), triggered in correspondence of the road trench upstream the route, that blocked the route. c) A shallow landslide triggered in a slope cultivated with vineyards, whose mobilized materials destroyed completely a road trait downstream.**

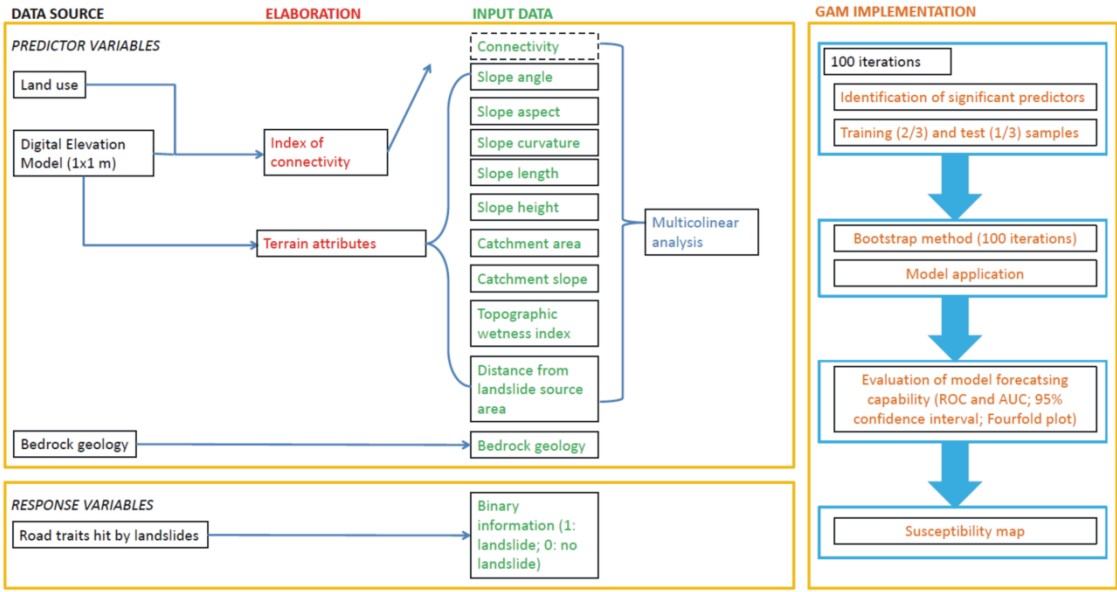

**Figure 4: Flow-chart containing the scheme for the implementation of the proposed model.**

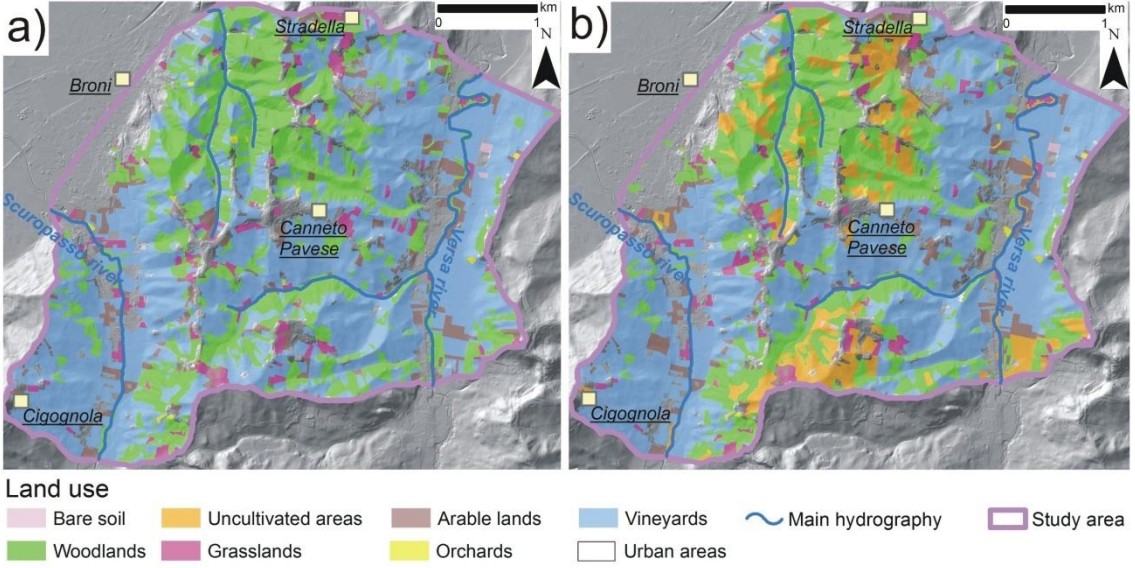

**Land use**

| | | | | | |
|---|---|---|---|---|---|
| Bare soil | Uncultivated areas | Arable lands | Vineyards | Main hydrography | Study area |
| Woodlands | Grasslands | Orchards | Urban areas | | |

**Figure 5: Potential land use scenarios used for the assessment of road susceptibility to shallow landslides in the study area, together with 1980 land use distribution (Fig. 2c): a) Scenario 3, correspondent to the transformation of actual uncultivated areas in woodlands; b) Scenario 4, correspondent to an increase in abandoned areas similar to that one in 1980-2015 period.**

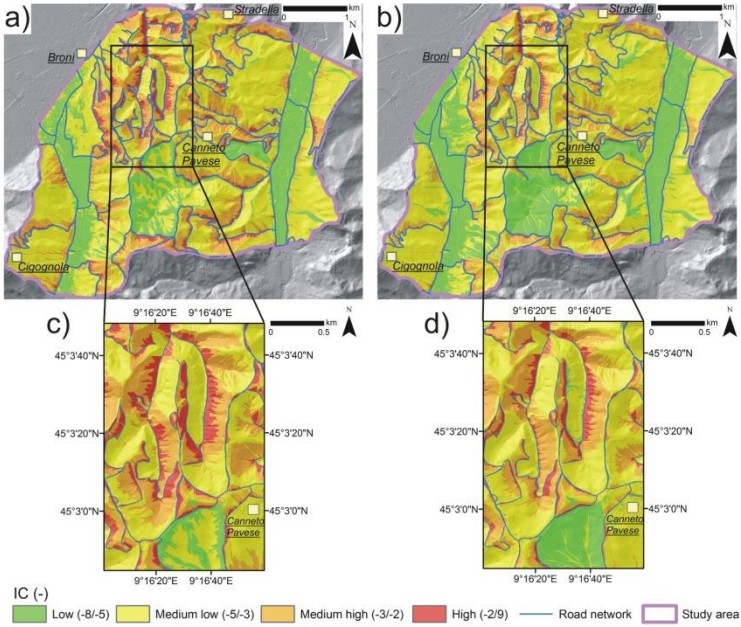

IC (-)

| | | | | | |
|---|---|---|---|---|---|
| Low (-8/-5) | Medium low (-5/-3) | Medium high (-3/-2) | High (-2/9) | Road network | Study area |

**Figure 6: Actual (2015) IC maps corresponding to the linear calculation of the index since $W_{lin}$ (a) and to the non-linear calculation of the index since $W_{nl}$ (b). A detail of the northern sector of the study areas is reported for $IC_{lin}$ (c) and $IC_{nl}$ (d) maps.**

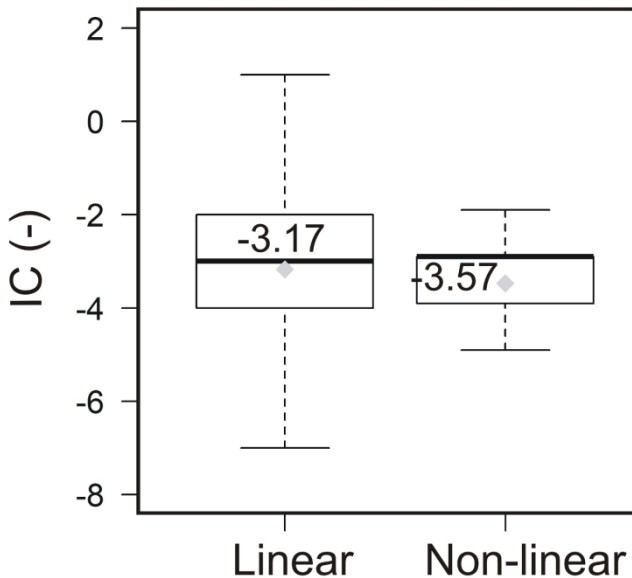

**Figure 7: Boxplot of IC values distribution for the actual scenario (2015), for the linear and non-linear calculation of the index.**

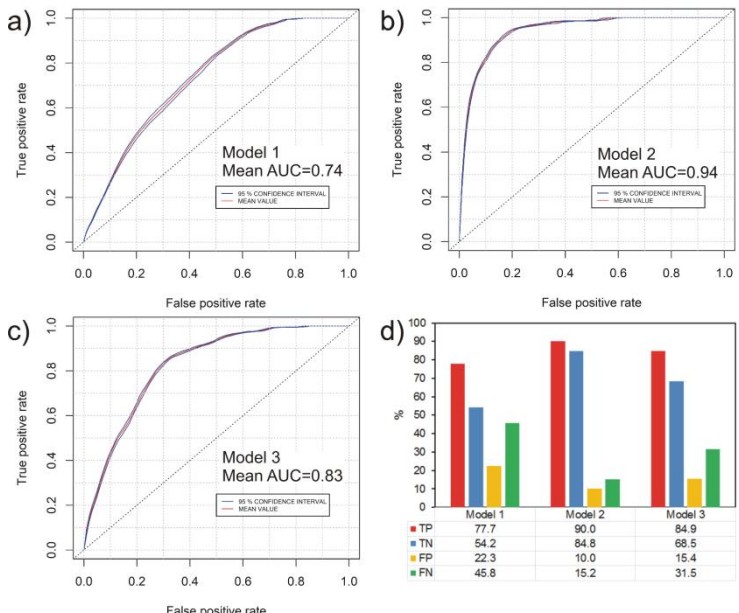

**Figure 8: 95% bootstrap confidence bands of ROCs: a) Model 1; b) Model 2; c) Model 3. d) Percentage of true positives (TP), true negatives (TN), false positives (FP), false negatives (FN) of the different models.**

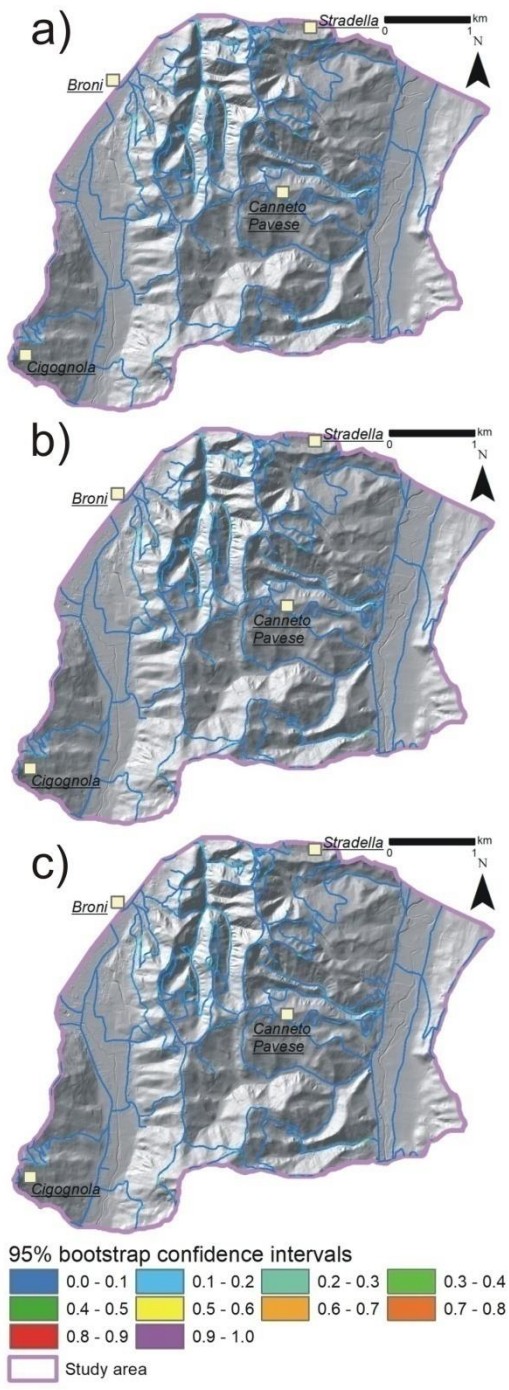

95% bootstrap confidence intervals

| | | | |
|---|---|---|---|
| 0.0 - 0.1 | 0.1 - 0.2 | 0.2 - 0.3 | 0.3 - 0.4 |
| 0.4 - 0.5 | 0.5 - 0.6 | 0.6 - 0.7 | 0.7 - 0.8 |
| 0.8 - 0.9 | 0.9 - 1.0 | | |

Study area

**Figure 9: Maps of the amplitude of 95% bootstrap confidence intervals of the probability associated to each pixel of the studied area: a) Model 1; b) Model 2; c) Model 3.**

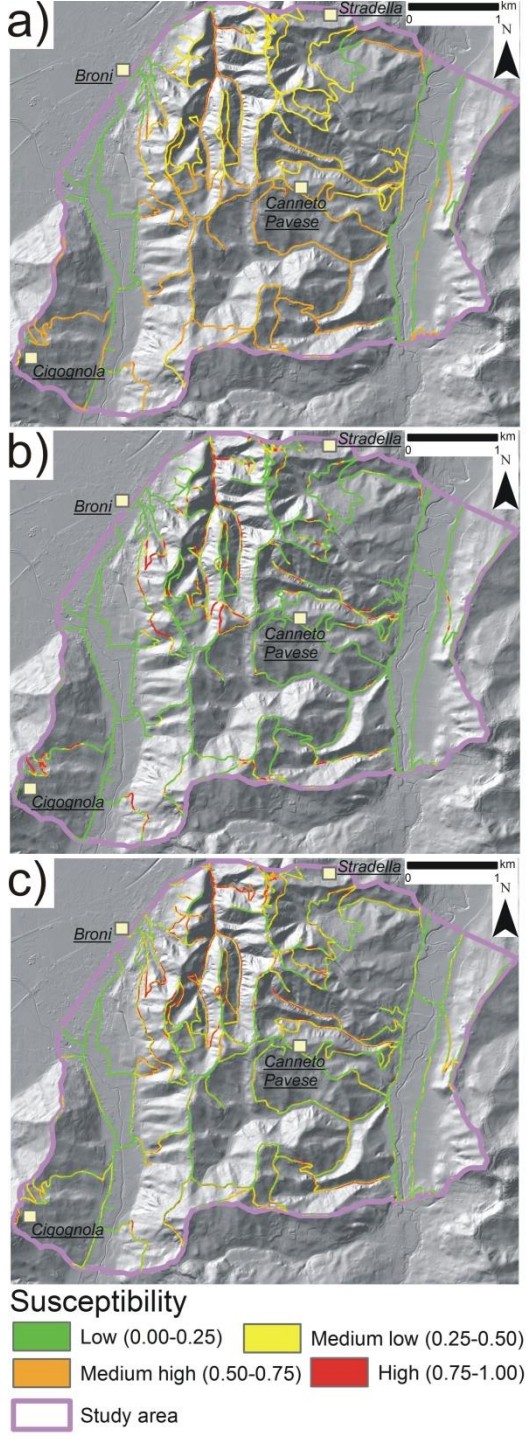

## Susceptibility

| | | | |
|---|---|---|---|
| 🟩 | Low (0.00-0.25) | 🟨 | Medium low (0.25-0.50) |
| 🟧 | Medium high (0.50-0.75) | 🟥 | High (0.75-1.00) |
| ▭ | Study area | | |

**Figure 10: Maps of the susceptibility of the road segments to be affected by shallow landslides: a) Model 1; b) Model 2; c) Model 3.**

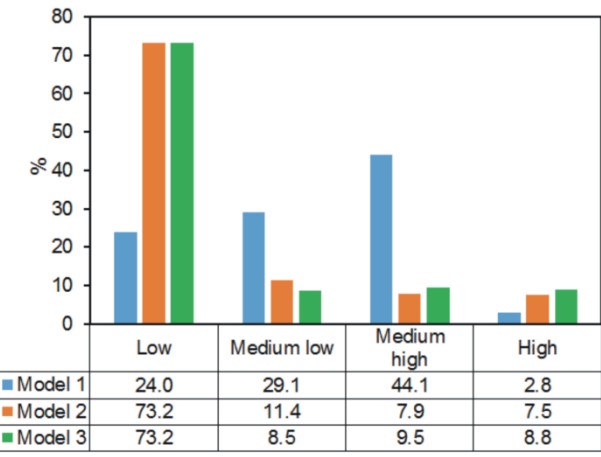

| | Low | Medium low | Medium high | High |
|---|---|---|---|---|
| Model 1 | 24.0 | 29.1 | 44.1 | 2.8 |
| Model 2 | 73.2 | 11.4 | 7.9 | 7.5 |
| Model 3 | 73.2 | 8.5 | 9.5 | 8.8 |

**Figure 11: Percentage of the road network classified with low, medium-low, medium-high or high susceptibility to be affected by shallow landslides for each GAM model.**

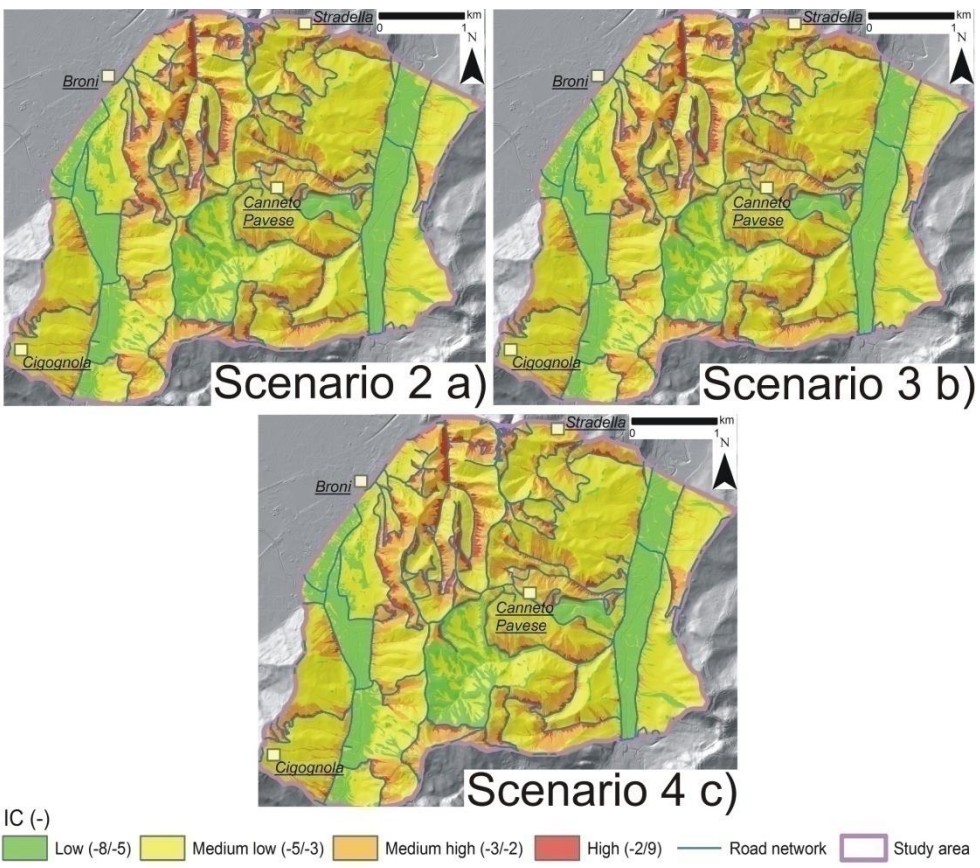

IC (-)
■ Low (-8/-5)   ■ Medium low (-5/-3)   ■ Medium high (-3/-2)   ■ High (-2/9)   — Road network   □ Study area

5      **Figure 12: IC$_{lin}$ maps of the different land use scenarios considered: a) Scenario 2: land use distribution equal to that one of 1980 (highest extension of vineyards); b) Scenario 3: transformation of actual uncultivated areas in woodlands; c) Scenario 4: increase in abandoned areas similar to that one occurred in 1980-2015 period.**

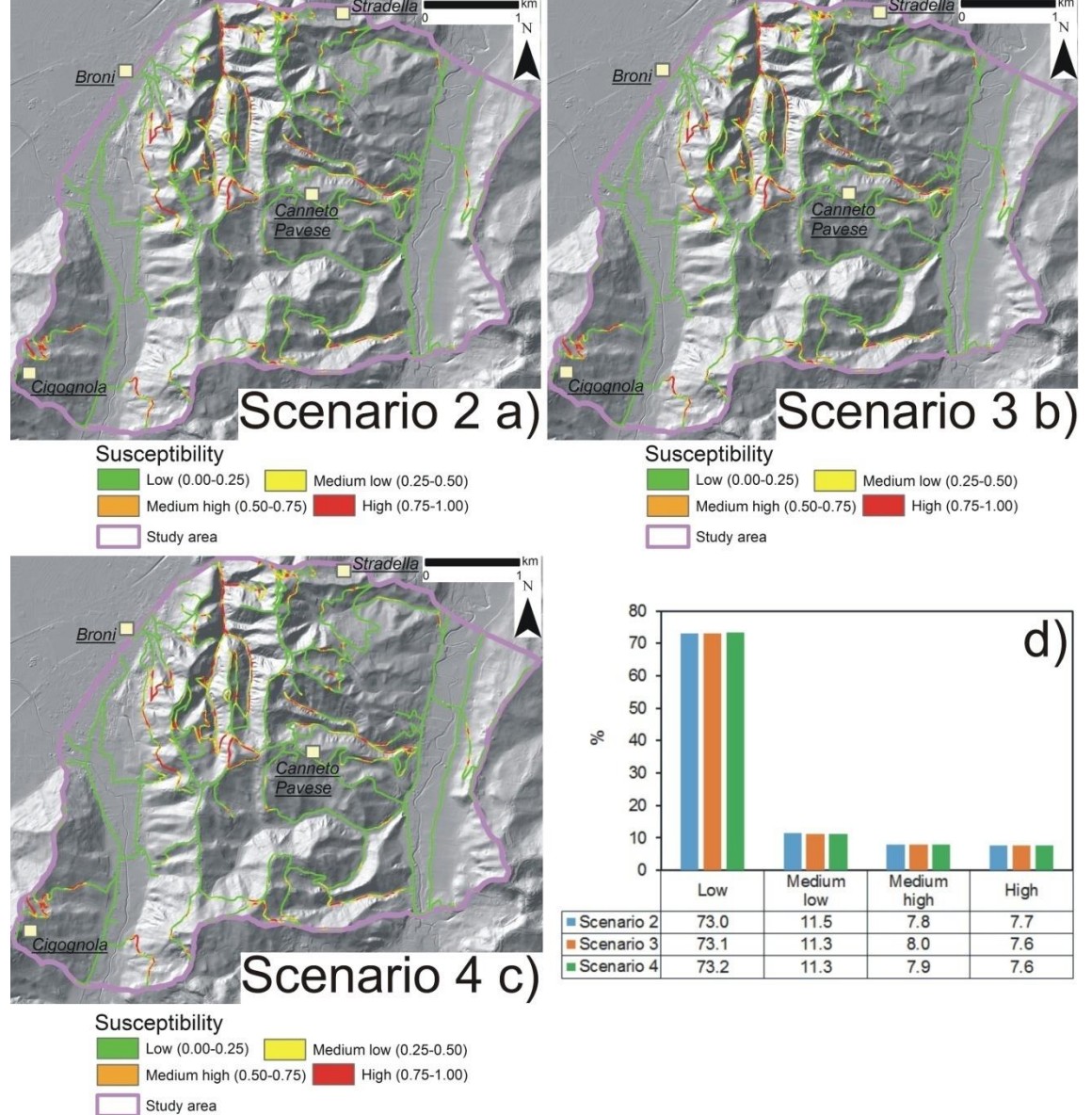

**Figure 13: Maps of the susceptibility of the road segments to be affected by shallow landslides according to the different land use scenarios: a) Scenario 2: land use distribution equal to that one of 1980 (highest extension of vineyards); b) Scenario 3: transformation of actual uncultivated areas in woodlands; c) Scenario 4: increase in abandoned areas similar to that one occurred in 1980-2015 period; d) percentage of the road network traits of different susceptibility classes for the considered scenarios.**

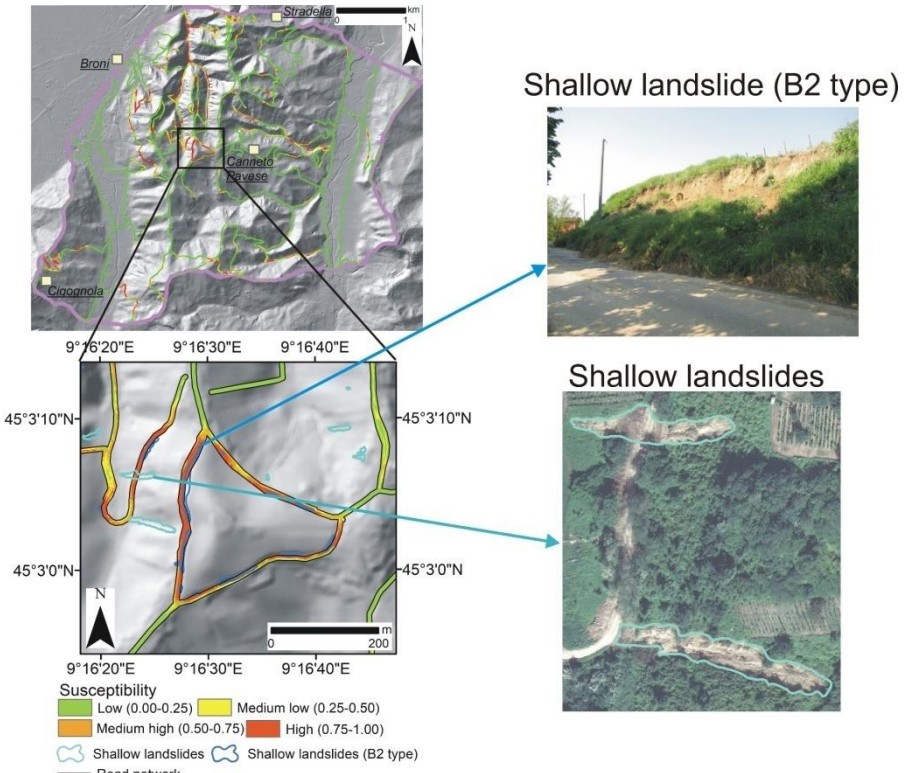

**Figure 14: Examples of correct assessment of the susceptibility performed by Model 2, for road sectors hit by B2 type shallow landslides and by other types of phenomena.**

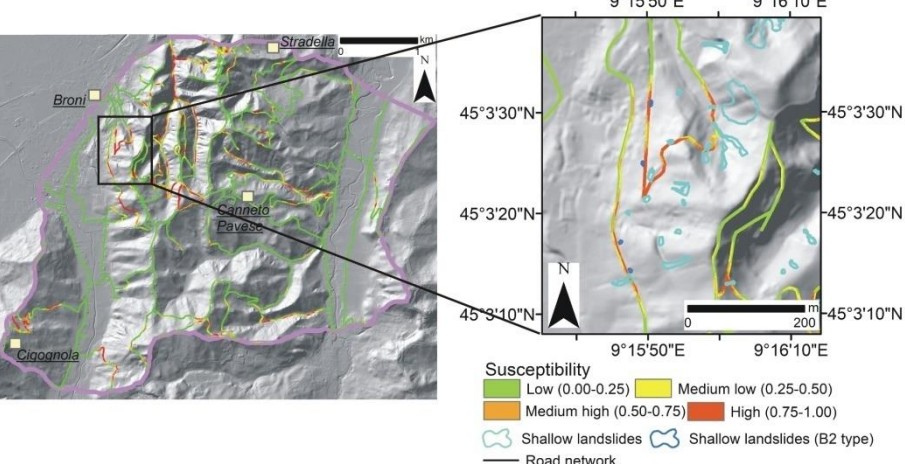

5  **Figure 15: False positive (FP) cases identified through the susceptibility map obtained from Model 2. FP cases are mostly located close (in a range lower than 250 m) to road sectors already affected by shallow landslides, in similar morphological and connectivity settings.**