# Peer review of "Estimation of the susceptibility of a road network to shallow landslides with the integration of the sediment connectivity"

_Natural Hazards and Earth System Sciences, 2017_

## Referee Comment (RC1) · Anonymous Referee #1 · 27 Feb 2018

The work is an interesting contribution to the journal, and it provides new insights on the relationship between roads and landslides, from a land management point of view. I have, however, some major concerns that should be addressed before the paper is ready for publication. My concerns are mostly related to the model construction and evaluation. According to the manuscript, the variables of the model were selected using the AIC criterion, but the authors do not explain which one: the backward approach, the forward one or the backward-forward? In the Forward method, one starts with an empty model, and iterate over all features. For each feature, the model is trained, and one select the feature which yields the best model according to a specific metric. Similarly, further features that yield the best improvement when combined with the

already selected ones are added. In the backward method we start with all features, and iteratively remove that one whose removal least hurt the performance, or leads to the biggest improvement. Therefore, the models selected by forward selection or backwards elimination might not be the same, even using the same model selection criterion. Also, the authors do not specify what criterion is considered to define the 'best' model achieved when adding/removing a feature. They only speak about the final performance of the model but do not provide any comparison between results obtained by adding or removing variables.

A further question arises: what's the reason behind choosing 80% as the threshold for variable acceptance? There is no justification for this choice, aside from an author pref-erence. While I do understand that 80% is a high number, what is the difference in the quality of the results at the change of this threshold? The authors should consider this a bit more in detail [i.e. as for the previous point, does removing/adding one variable or the other improve the results significantly? What if we select variables chosen more than 50% or 90% of the times?]. Addressing these two points would also improve the discussion in Chapt .4.2.1.

Another point is that currently there are no rational formulations for the indices that are kept or removed, other than the fact that they are a mathematical construct. What I mean is: is there a physical meaning behind the rejection or acceptance of such parameters? The description of the IC for the area helps to interpret its importance in the model, and the reason behind the increased quality of models that do include it in one way or another. However, the authors should also describe the other indices about the road network in their study (not just as a general statement on why they are important, as done in Chapt. 3.1.1), to justify their choice or confirm the model assumption. This would also help 'balance' the paper more: as of now, the focus on connectivity seems unbalanced, and similar to the previous work by (Persichillo, Bordoni, Cavalli, Crema, & Meisina, 2018).

Some minor comments arose as I read the manuscript.

[Figure]

English needs polishing. Some parts are too 'colloquial' (e.g. '' It is also worth noting that") or have some English mistakes, mostly in the first part of the manuscript e.g. "the evaluation of the importance of considering or neglecting sediment connectivity" is redundant, you can simply state 'the importance of considering sediment connectivity'. Line 31 p 3: "in the routes distribution that could be affected by shallow landslides" > is the distribution affected by shallow landslides or are the roads affected by it? Line 29 p 4. "The road sectors were built in correspondence of the valley floors or hillside, cutting a portion of a hillslope in correspondence of its medium part realising a halfway road" > this sentence is not clear, what Is a halfway road? Medium part of what, of the hillslope? Line 6 p 5: '30% of these shallow landslides WERE triggered in vineyards,' Line 10 p 5. What is a b2 type? Line 19 p 9 'to discriminate affected or not road sectors' this is redundant. It is clear that by discriminating affected road sections, it would remove those not affected.

Abstract needs rewording. Some concept are introduced without the reader knowing what they are, e.g. 'The random partition of the dataset used for building the model in two parts (training and test subsets), within a 100-fold bootstrap procedure.'

Literature in the introduction could be improved, e.g. about road networks and landslides (Bíl, Andrásik, Kubecek, Krivánková, & Vodák, 2017; Donnini et al., 2017; Hearn, Hunt, Aubert, & Howell, 2008; Martinović, Gavin, Reale, & Mangan, 2018; Penna, Borga, Aronica, Brigandì, & Tarolli, 2014; Postance, Hillier, Dijkstra, & Dixon, 2017; P. Tarolli, Calligaro, Cazorzi, & Dalla Fontana, 2013; Paolo Tarolli & Sofia, 2016) and about road-landslides and climate changes (Klose, Auerbach, Herrmann, Kumerics, & Gratzki, 2017; Michaelides, 2014; Strauch, Raymond, Rochefort, Hamlet, & Lauver, 2015)

Line 30 p 5: slope aspect (ASP), slope curvature (CURV) > these would be better defined as simply aspect and curvature. Also, what is the 'slope height'? Line 10 p 6. Why using the multiflow algorithm? Wouldn't it be more consistent to use the D-Infinity since the sediment connectivity is also computed through D-inf, which is more accurate and

less dispersive, especially on hillslopes? Line 15 p 6: why not considering a geodesic distance or a 3d distance, rather than a simply Euclidean distance? I'd assume that on a hilly slope, a 3d distance might be very different from a Euclidean one. Also, was this distance evaluated considering possible flow direction? I would assume that the possible direction/movement of a landslide would follow topography, and more specifically a shortest topographic travelling distance, rather than a simple 2d distance to the road network. Thus a 3d topographic distance might be more appropriate as a vulnerability index. Also in this paragraph, 'lowest distance' should be 'shortest distance.' Line 15 p 9: is there a reference for this holdout bootstrap method? Line 2 p 10: a buffer of 5 m from the middle of each road sector > what's the reasoning behind this buffer? Is this in line with the road size? Should it be varied considering main roads or minor roads?

The first paragraph of the discussion is not needed. It is a repetition of the introduction. Line 10 p 14. The authors state '' Instead, the proposed approach helps in filling the gaps and the limits still open in the definition of a reliable and, potentially, repeatable methodology.". However, I do not see how this was demonstrated. The method was replicated in their study case, so it is not that different from the previous literature they mentioned, where the methodologies were "developed and tested for particular geological/geomorphological settings." Line 13 to 19> this is not about the current work. If anything, this should be mentioned when the authors justify the choice of the data-driven method, but it is not a result to discuss. Line 7 to 15 p 15> this again is not about the current work. It should be eventually mentioned when the authors justify the choice of the IC, or to highlight similarities between their results and the mentioned works, which is not the case currently. Line 15 p 16 "They are in a buffer of less than 250 m, in particular between 50 and 200 m, respect to sectors hit in past, and they present morphological and connectivity features similar to threatened traits." > shouldn't the authors also include these locations (sectors hit in the past) in their assessment as reference data? If their model is meant to be feasible outside their study area and not the case-specific, it should be able to identify correctly all the elements, not only those triggered in one specific event. The first paragraph of line 17

[Figure]

> 'Hence, more detailed scenarios of susceptibility changes about land use changes will take into account also for the morphological modifications linked to these changes, also using a higher resolution DEM (less than 1 m).' is this a future research line or a result?

references

Bíl, M., Andrásik, R., Kubecek, J., Krivánková, Z., & Vodák, R. (2017). RUPOK: An online landslide risk tool for road networks. In Advancing Culture of Living with Landslides (Vol. 5, pp. 19–26). https://doi.org/10.1007/978-3-319-53483-1_4 Donnini, M., Napolitano, E., Salvati, P., Ardizzone, F., Bucci, F., Fiorucci, F., . . . Guzzetti, F. (2017). Impact of event landslides on road networks: a statistical analysis of two Italian case studies. Landslides, 14(4), 1521–1535. https://doi.org/10.1007/s10346-017-0829-4 Hearn, G., Hunt, T., Aubert, J., & Howell, J. (2008). Landslide impacts on the road network of Lao PDR and the feasibility of implementing a slope management programme. International Conference on . . .. Retrieved from https://assets.publishing.service.gov.uk/media/57a08ba8ed915d622c000e03/Seacp21-02.pdf Klose, M., Auerbach, M., Herrmann, C., Kumerics, C., & Gratzki, A. (2017). Landslide Hazards and Climate Change Adaptation of Transport Infrastructures in Germany. In Advancing Culture of Living with Landslides (pp. 535–541). Cham: Springer International Publishing. https://doi.org/10.1007/978-3-319-59469-9_48 Martinović, K., Gavin, K., Reale, C., & Mangan, C. (2018). Rainfall thresholds as a landslide indicator for engineered slopes on the Irish Rail network. Geomorphology. https://doi.org/10.1016/j.geomorph.2018.01.006 Michaelides, S. (2014). Vulnerability of transportation to extreme weather and climate change. Natural Hazards, 72(1), 1–4. https://doi.org/10.1007/s11069-013-0975-5 Penna, D., Borga, M., Aronica, G. T., Brigandì, G., & Tarolli, P. (2014). The influence of grid resolution on the prediction of natural and road-related shallow landslides. Hydrology and Earth System Sciences, 18(6), 2127–2139. https://doi.org/10.5194/hess-18-2127-2014 Persichillo, M. G., Bordoni, M., Cavalli, M., Crema, S., & Meisina, C. (2018). The

role of human activities on sediment connectivity of shallow landslides. CATENA, 160, 261–274. https://doi.org/10.1016/J.CATENA.2017.09.025 Postance, B., Hillier, J., Dijkstra, T., & Dixon, N. (2017). Extending natural hazard impacts: an assessment of landslide disruptions on a national road transportation network. Environmental Research Letters, 12(1), 14010. https://doi.org/10.1088/1748-9326/aa5555 Strauch, R. L., Raymond, C. L., Rochefort, R. M., Hamlet, A. F., & Lauver, C. (2015). Adapting transportation to climate change on federal lands in Washington State, U.S.A. Climatic Change, 130(2), 185–199. https://doi.org/10.1007/s10584-015-1357-7 Tarolli, P., Calligaro, S., Cazorzi, F., & Dalla Fontana, G. (2013). Recognition of surface flow processes influenced by roads and trails in mountain areas using high-resolution topography. EUROPEAN JOURNAL OF REMOTE SENSING, 46, 176–197. https://doi.org/10.5721/EuJRS20134610 Tarolli, P., & Sofia, G. (2016). Human topographic signatures and derived geomorphic processes across landscapes. Geomorphology, 255, 140–161. https://doi.org/10.1016/j.geomorph.2015.12.007

---

## Referee Comment (RC2) · Anonymous Referee #2 · 25 Mar 2018

Dear Editor, The paper Estimation of the susceptibility of a road network to shallow landslides with the integration of the sediment connectivity" by Bordoni et al. present an interesting study case for landslide susceptibility applied to roads in a small area of north Apennines, Italy. One of the main contribution (as the authors state in the title) is the integration of sediment connectivity in the statistical model, using different land use scenarios. From my point of view, the quality of the paper, the method used, the results and the discussion make the paper suitable for publication in NHESS journal. I suggest some highlights of this kind of studies in the introduction, discussions and conclusion parts of the paper. In my opinion, some minor issues must be solved before the publication: (i) please provide more details in "the study area" section about the

method used for land use mapping; also, you can move here some information about landslide inventory - data acquisition; (ii) the discussion part must rewrote, at least the first 4 paragraphs; (iii) please change the Fig. 9, in order to make it representative according to the associated legend; and (iv) try to find other color for the roads (at least in the Figures 1, 3, and 9) for increasing the contrast and readability of the map. Other minor problems you will find in the *.pdf file attached.

Please also note the supplement to this comment:
https://www.nat-hazards-earth-syst-sci-discuss.net/nhess-2017-457/nhess-2017-457-RC2-supplement.pdf
* * *
[Figure]

**Supplement:**

[revised manuscript text omitted]

---

## Author Comment (AC1) · 26 Mar 2018

The authors are grateful to the Anonymous Referee #1, whose comments and suggestions will contribute towards an improvement of the final paper. We will also consider the suggestions proposed by the Anonymous Referee #2. Point-by-point replies to the Anonymous Referee #1's comments follow.

Comment 1 The work is an interesting contribution to the journal, and it provides new insights on the relationship between roads and landslides, from a land management point of view. I have, however, some major concerns that should be addressed before the paper is ready for publication.

[Figure]

Response to Comment 1 We thank the Referee for the appreciation of our work. In revising the paper, we considered all the suggestions and comments, which allowed to clarify several aspects and to improve the article overall quality.

Comment 2 According to the manuscript, the variables of the model were selected using the AIC criterion, but the authors do not explain which one: the backward approach, the forward one or the backward-forward? In the Forward method, one starts with an empty model, and iterate over all features. For each feature, the model is trained, and one select the feature which yields the best model according to a specific metric. Similarly, further features that yield the best improvement when combined with the already selected ones are added. In the backward method we start with all features, and iteratively remove that one whose removal least hurt the performance, or leads to the biggest improvement. Therefore, the models selected by forward selection or backwards elimination might not be the same, even using the same model selection criterion. Also, the authors do not specify what criterion is considered to define the 'best' model achieved when adding/removing a feature. They only speak about the final performance of the model but do not provide any comparison between results obtained by adding or removing variables.

Response to Comment 2 We clarified this aspect about the implementation of the GAM model for road susceptibility estimation. For the selection of the explanatory variables, we used the 'step.Gam' command of the R package 'gam'. This command, as explained in the library manual (available at URL https://cran.r-project.org/web/packages/gam/gam.pdf), allows to choose any of the three approaches. We choose to select the variables allowing both directions in the step-wise search, using the option direction="both" in issuing the step.Gam command. The selected "best" model is the one that minimizes the Akaike Iteration Criterion statistic. This procedure was repeated 100 times using 100 bootstrap extractions form the same dataset. The final model was chosen according to an acceptance threshold of 80%. In the response to comment 3, it is present a reasonable justification of this threshold.

Comment 3 A further question arises: what's the reason behind choosing 80% as the threshold for variable acceptance? There is no justification for this choice, aside from an author preference. While I do understand that 80% is a high number, what is the difference in the quality of the results at the change of this threshold? The authors should consider this a bit more in detail [i.e. as for the previous point, does removing/adding one variable or the other improve the results significantly? What if we select variables chosen more than 50% or 90% of the times?]. Addressing these two points would also improve the discussion in Chapt .4.2.1.

Response to Comment 3 As suggested by the Referee in his Comment 2, we performed a sensitivity analysis to assess the role of each predictor variable on the accuracy of the GAM models. This analysis allowed also evaluating the change in predictive accuracy related to adding or removing a set of predictors according to a threshold of selection different than the used 80% or related to adding or removing a particular predictor. It is important to highlight that the results of this sensitivity analysis shown in the paper referred to the susceptibility model which had the best predictive accuracy, that is Model 2 considering all the predictors selected using the threshold of 80% and the index of connectivity calculated in a linear way. Instead, the quantitative changes on the predictive accuracy related to different sets of predictors were similar also considering Model 1 (all the predictors selected using the threshold of 80% without considering the index of connectivity) and Model 3 (all the predictors selected using the threshold of 80% and the index of connectivity calculated in the non-linear way). Table 1 attached to these responses showed the results of this sensitivity analysis. First, the effects on the model accuracy related to a change of the value of selection threshold used for choosing the predictor variables used for the creation of the final susceptibility model were evaluated. According to the percentages of selection of each variable in the 100-fold bootstrap procedure (Tab. 2 of the manuscript), also thresholds of 50% and 90% of selection frequency were considered and compared to the used threshold of 80%. A threshold of selection frequency lower than 50% was not considered significant. Considering a threshold equal to 50%, also CA (chosen as a linear variable) had to be
inserted for modeling the susceptibility. Instead, the mean predictive accuracy of the model, estimated in terms of AUC values, did not change, for both the training sets, the test sets and the final model. The difference in the predictive accuracy was lower than 0.01. Instead, concerning a threshold equal to 90%, CURV, HEI and TWI had to be removed, because their frequency selections were between 85 and 88%. In this case, the mean predictive accuracy of the best model (Model 2) decreased from 0.90 to 0.84 and from 0.94 to 0.88 for training/test sets and for the final models, respectively. Removing a predictor or a set of these from the susceptibility model caused a decrease of the accuracy due to a reduction in explaining the physical relations between the predisposing factors and the resulting effects on the response variable, in this case represented by the road sectors hit by shallow landslides. These results demonstrated that a threshold of selection of the predictors equal to 80% allowed to obtain the sets of predisposing factors able to estimate in the best reliable and effective way the susceptibility of the road network to be affected by shallow landslides. Furthermore, a sensitivity analysis of the different predictors considered as predisposing factors for road susceptibility was performed. This analysis consisted in running one of the models created considering a threshold of selection frequency equal to 80% (Model 1, 2, 3), removing each time one of the selected predictors or adding each time one of the other predictors, whose frequency of selection was lower than 80%. In this way, the sensitivity of the model to each predictor could be quantified. Starting from Model 2, which was the best in terms of reliability, the removal of a particular predictor could affect the accuracy. Removing SL or DIST caused a reduction of the predictive accuracy, for both training sets, test sets and final models, of 0.15-0.16. Instead, this reduction was lower than the one quantified if in the model IC was not taken into account (Model 1). In fact, the absence of IC provoked a decrease in the accuracy of 0.19-0.20. The removal of CS caused a moderate reduction of the accuracy, correspondent to 0.11. While, removing one of the other chosen parameters (CURV, HEI, TWI, GEO) provoked only a slight decrease in the predictive accuracy, in the order of 0.02-0.06. Moreover, adding alternatively to the chosen predictors one of the other predisposing factors (CA, ASP, LEN) did not

modify significantly the reliability of the models. The predictive accuracy improved at most 0.01 for both training sets, test sets and final models. These results confirmed the significant sensitivity of the susceptibility model to IC, especially the one estimated in a linear way. Neglecting IC in these models caused a big decrease in the effectiveness, which affects significantly the susceptibility classification of the road network. Furthermore, SL and DIST also affected significantly the accuracy of the final susceptibility model and had to be considered for obtaining a correct classification of road network. The models were more slightly sensitive to the other chosen predictors (CURV, HEI, TWI, GEO). Instead, the leakage of only one of those parameters could decrease the final reliability of the road susceptibility. The models were not sensitive to the other considered predisposing factors (CA, ASP, LEN). Thus, these parameters did not allow for a further improvement of the susceptibility models reliability and could be correctly excluded from the models. These results confirmed further the goodness of choosing a threshold of selection frequency of the predictors equal to 80%. It is important to note that the standard deviation of accuracy on training and test sets was of 0.01 for all the models, while the range of the 95 % confidence interval of AUC was of 0.02 for all the models.

Comment 4 Another point is that currently there are no rational formulations for the indices that are kept or removed, other than the fact that they are a mathematical construct. What I mean is: is there a physical meaning behind the rejection or acceptance of such parameters? The description of the IC for the area helps to interpret its importance in the model, and the reason behind the increased quality of models that do include it in one way or another. However, the authors should also describe the other indices about the road network in their study (not just as a general statement on why they are important, as done in Chapt. 3.1.1), to justify their choice or confirm the model assumption. This would also help 'balance' the paper more: as of now, the focus on connectivity seems unbalanced, and similar to the previous work by (Persichillo, Bordoni, Cavalli, Crema, & Meisina, 2018).

Response to Comment 4 As done in the manuscript for IC feature, we better analyzed the distribution of the other predictor variables considered for building the susceptibility models of the study area. This allowed integrating the analysis of the role played by IC in road susceptibility, giving indications also on the role played by the other features. The maps of the distribution of some predictors (SL, ASP, CURV, LEN, HEI, CA, CS, TWI, DIST) were present in Fig. 1 attached to these responses. The distribution of the bedrock geological formations (GEO) in the study area was already shown in Fig. 1 of the manuscript. The procedure adopted for the selection of the most significant variables used for the creation of the susceptibility models excluded ASP, LEN and CA parameters. Distribution of ASP in the study area (Fig. 1b) showed that the exposition of the slopes close to the roads was very variable, without the identification of peculiar features. While, LEN (Fig. 1d) and CA (Fig. 1f) values close to the road sectors were in a quite narrow range, between 2 and 150 m and around 102 m2, respectively. The particular distributions of these parameters confirmed their not significant roles in the evaluation of the road susceptibility. Thus, they could be correctly not considered in GAM models. Concerning the predictors selected by the 100-fold bootstrap procedure, roads were located especially close to hillslopes of medium-high SL (higher than 10°, except for the routes located in the floors of the river valleys; Fig. 1a), limited HEI (lower than 50 m; Fig. 1e) and with shallow landslides triggering zones located very close to the road network (lower than 150 m; Fig. 1i). In fact, the affected road sectors were generally road segments downstream to slopes characterized by high slope gradient (> 20°), limited height (< 50 m) and with shallow landslides triggering zones located very close to the road network (40-100 m). These sectors were correctly classified as susceptible by the best model (Model 2). Also CS had an important effect on the susceptibility of a road to be hit by shallow failures. Roads were located close to hillslopes with very low (0-5°) or very high (20-31°) CS values (Fig. 1g). Affected roads, classified as susceptible by the implemented models, corresponded to road traits located in correspondence of very high values of CS, generally between 20-28°. As highlighted in the Response to Comment 3, CURV, TWI and GEO were selected by the

methodology as significant predictors, but they had a lower effect on the accuracy of the models. This meant that they explained less than the other selected predictors the susceptibility of a road to shallow landslides. Instead, CURV values (Fig. 1c) close to the roads were generally slightly negative (lower than -0.05) and the affected sectors were in correspondence of the lowest CURV values (around -0.40). TWI (Fig. 1h) was generally positive in correspondence of road traits, with values higher than 5 close to sectors affected by shallow landslides. Moreover, damaged road traits were mainly located in areas where GEO was composed of medium low-permeable arenaceous conglomeratic materials (Monte Arzolo Sandstones, Rocca Ticozzi Conglomerates) or impermeable silty-sandy marly bedrock (Montù Beccaria Formation, Sant'Agata Fossili Marls).

Comment 5 Some minor comments arose as I read the manuscript. English needs polishing. Some parts are too 'colloquial' (e.g. '' It is also worth noting that") or have some English mistakes, mostly in the first part of the manuscript e.g. "the evaluation of the importance of considering or neglecting sediment connectivity" is redundant, you can simply state 'the importance of considering sediment connectivity'.

Response to Comment 5 We thank the Referee for this suggestion. We carefully performed a detailed revision of English to clarify several unclear sections, to delete colloquial sentences and mistakes and to improve the overall quality of the manuscript. We also considered the suggested correction of the Referee, changing the sentence "...the evaluation of the importance of considering or neglecting sediment connectivity..." in "...the importance of considering connectivity...".

Comment 6 Line 31 p 3: "in the routes distribution that could be affected by shallow landslides" > is the distribution affected by shallow landslides or are the roads affected by it?

Response to Comment 6 We rearranged this sentence to clarify the expressed concept. We meant that the road sectors of a particular area, prone to shallow landslides,

could be affected by phenomena triggered in the closest slopes, which could damage the roads themselves. Thus, we modified this sentence in: "...the road sectors potentially affected by shallow landslides...".

Comment 7 Line 29 p 4. "The road sectors were built in correspondence of the valley floors or hillside, cutting a portion of a hillslope in correspondence of its medium part realising a halfway road" > this sentence is not clear, what Is a halfway road? Medium part of what, of the hillslope?

Response to Comment 7 We rearranged this sentence to clarify the expressed concept. In the study area, the roads were built in correspondence of the valley floors or in the medium part of a hillslope, cutting its continuity. This sentence was then modified in: "...The roads were built in correspondence of the valley floors or in the medium part of a hillslope, cutting its continuity...".

Comment 8 Line 6 p 5: '30% of these shallow landslides WERE triggered in vineyards

Response to Comment 8 We added "were" where it leaked, to correct this grammatical error.

Comment 9 Line 10 p 5. What is a b2 type?

Response to Comment 9 In the set of the shallow landslides hitting roads of the study area, 24 failures (5% of the total number) were roto-translational slides affecting the trench of a cut realized for building the road itself. Zizioli et al. (2013) and Persichillo et al. (2018) named these failures as "B2" type, thus we decided to recall this type of landslides with the same term. We clarified this concept with the following sentence: "...Moreover, 24 failures (5% of the total number) were roto-translational slides affecting the trench of a cut realized for building a road. These phenomena were named as B2 type, according to the term used by Zizioli et al. (2013) and Persichillo et al. (2018)...". References: Persichillo et al. (2018): Persichillo, M. G., Bordoni, M., Cavalli, M., Crema, S. and Meisina, C.: The role of human activities on sediment connectivity

of shallow landslides, Catena 160, 261-274, doi:10.1016/j.catena.2017.09.025, 2018. Zizioli et al. (2013): Zizioli, D., Meisina, C., Valentino, R., and Montrasio, L.: Comparison between different approaches to modeling shallow landslide susceptibility: a case history in Oltrepo Pavese, Northern Italy, Nat. Hazards Earth Syst. Sci., 13, 559–573, doi:10.5194/nhess-13-559-2013, 2013.

Comment 10 Line 19 p 9 'to discriminate affected or not road sectors' this is redundant. It is clear that by discriminating affected road sections, it would remove those not affected.

Response to Comment 10 We thank the Referee for this suggestion. We modified this sentence in: "...to discriminate affected road sectors...".

Comment 11 Abstract needs rewording. Some concept are introduced without the reader knowing what they are, e.g. 'The random partition of the dataset used for building the model in two parts (training and test subsets), within a 100-fold bootstrap procedure.'

Response to Comment 11 We thank the Referee for this revision. We modified the Abstract, in those unclear parts to introduce better several concepts not completely presented before. In particular, at pag. 1 lines 14-16, we modified the sentence in "...For these reasons, this paper aimed to develop and test a data-driven model for the identification of road sectors that are susceptible to be hit by shallow landslides triggered in slopes upstream to the infrastructure. This model was based on the Genetic Algorithm Method, where the function relating predictors and response variable is an empirically fitted smooth function that allows fitting the data in the more likely functional form, considering also non-linear relations...". Moreover, at pag. 1 lines 16-17, we modified the sentence in "...This work also analyzed the importance, on the estimation of the susceptibility, of considering or not the sediment connectivity, which influences the path and the travel distance of the materials mobilized by a slope failure till a potential barrier as a road...". At pag. 1 lines 18-22, we modified the sentence

in "...The most significant explanatory variables were selected by a random partition of the available dataset in two parts (training and test subsets), for 100 times according to a bootstrap procedure. These variables (selected 80 times at least by the bootstrap procedure) were used to build the final susceptibility model, whose accuracy was estimated through a 100-fold repetition of holdout method for regression based on the training and test sets created through the 100 bootstrap model selection..."

Comment 12 Literature in the introduction could be improved, e.g. about road networks and landslides (Bíl, Andrásik, Kubecek, Krivánková, & Vodák, 2017; Donnini et al., 2017; Hearn, Hunt, Aubert, & Howell, 2008; Martinoviᶦc, Gavin, Reale, & Mangan, 2018; Penna, Borga, Aronica, Brigandì, & Tarolli, 2014; Postance, Hillier, Dijkstra, & Dixon, 2017; P. Tarolli, Calligaro, Cazorzi, & Dalla Fontana, 2013; Paolo Tarolli & Sofia, 2016) and about road-landslides and climate changes (Klose, Auerbach, Herrmann, Kumerics, & Gratzki, 2017; Michaelides, 2014; Strauch, Raymond, Rochefort, Hamlet, & Lauver, 2015).

Response to Comment 12 We improved the literature review about the interactions between landslides and roads and about the effects of climate change in events of landslides triggering affecting road networks. In particular, we added in the manuscript the following references related to the interactions between landslides and roads: Bil et al. (2017): Bil, M., Andrasik, R., Kubecek, J., Krivankova, Z. and Vodak, R.: RUPOK: An online landslide risk tool for road networks, in: Advancing culture of living with landslides, edited by: Mikos, M., Vilimek, V., Yin, Y., and Sassa, K., Springer, Cham, 19-26, 2017. Donnini et al. (2017): Donnini, M., Napolitano, E., Salvati, P., Ardizzone, F., Bucci, F., Fiorucci, F., Santangelo, M., Cardinali, M. and Guzzetti F.: Impact of event landslides on road networks: a statistical analysis of two Italian case studies, Landslides 14, 4, 1521–1535, doi:10.1007/s10346-017-0829-4, 2017. Hearn et al. (2008): Hearn, G., Hunt, T., Aubert, J. and Howell, J.: Landslide impacts on the road network of Lao PDR and the feasibility of implementing a slope management programme, South East Asia Community Access Programme (SEACAP), Department for International Development, United Kingdom. Martinovic et al. (2018): Martinovic, K., Gavin, K., Reale, C. and Mangan, C.: Rainfall thresholds as a landslide indicator for engineered slopes on the Irish Rail network, Geomorphology 306, 40-50, doi:10.1016/j.geomorph.2018.01.006, 2018. Penna et al. (2014): Penna, D., Borga, M., Aronica, G. T., Brigandì, G. and Tarolli, P.: The influence of grid resolution on the prediction of natural and road-related shallow landslides. Hydrol. Earth Syst. Sci. 18, 6, 2127–2139, doi:10.5194/hess-18-2127-2014, 2014. Postance et al. (2017): Postance, B., Hillier, J., Dijkstra, T. and Dixon, N.: Extending natural hazard impacts: an assessment of landslide disruptions on a national road transportation network, Environ. Res. Let. 12, 1, 14010, doi:10.1088/1748-9326/aa5555, 2017. Tarolli et al. (2013): Tarolli, P., Calligaro, S., Cazorzi, F. and Dalla Fontana, G.: Recognition of surface flow processes influenced by roads and trails in mountain areas using high-resolution topography, Eur. J. Remote Sens. 46, 176–197, doi:10.5721/EuJRS20134610, 2013. Tarolli and Sofia (2016): Tarolli, P. and Sofia, G.: Human topographic signatures and derived geomorphic processes across landscapes, Geomorphology 255, 140–161, doi:10.1016/j.geomorph.2015.12.007, 2016. Winter et al. (2016): Winter, M. G., Shearer, B., Palmer, D., Peeling, D., Harmer, C. and Sharpe J.: The economic impact of landslides and floods on the road network, Procedia Eng. 143, 1425-1434, doi:10.1016/j.proeng.2016.06.168, 2016. Moreover, we added the following references related to the effects of climate change in triggering events of landslides affecting road networks: Klose et al. (2017): Klose, M., Auerbach, M., Herrmann, C., Kumerics, C. and Gratzki, A.: Landslide hazards and climate change adaptation of transport infrastructures in Germany, in: Advancing culture of living with landslides, edited by: Mikos, M., Vilimek, V., Yin, Y., and Sassa, K., Springer, Cham, 535-541, 2017. Michaelides (2014): Michaelides, S.: Vulnerability of transportation to extreme weather and climate change, Nat. Hazards 72, 1, 1–4, doi:10.1007/s11069-013-0975-5, 2014. Strauch, R. L., Raymond, C. L., Rochefort, R. M., Hamlet, A. F. and Lauver, C.: Adapting transportation to climate change on federal lands in Washington State, U.S.A, Clim. Change 130, 2, 185–199, doi:10.1007/s10584-015-1357-7, 2015.

Comment 13 Line 30 p 5: slope aspect (ASP), slope curvature (CURV) > these would be better defined as simply aspect and curvature.

Response to Comment 13 We thank the Referee for this suggestion. We defined these two geomorphological parameters only as "aspect" and "curvature", respectively.

Comment 14 Also, what is the 'slope height'?

Response to Comment 14 Slope height represents the elevation difference between the source area of a shallow landslide and the bottom of the hillslope where this failure occurred. For clarifying this concept, we added the following sentence: "...Slope height (HEI) represented the elevation difference between the source area of a shallow landslide and the bottom of the hillslope where this failure occurred".

Comment 15 Line 10 p 6. Why using the multiflow algorithm? Wouldn't it be more consistent to use the D-Infinity since the sediment connectivity is also computed through D-inf, which is more accurate and less dispersive, especially on hillslopes?

Response to Comment 15 Multiflow direction algorithm was used for the computation of the catchment area and the catchment slope. Catchment area and catchment slope were used as proxies for soil moisture and soil depth and for the destabilizing forces upstream that can provoke the development of a landslide, respectively. Multiflow direction algorithm distributed the water flow to all neighboring downslope cells weighted according to slope angle, avoiding the flow concentration to particular lines sometimes unrealistic. In the case of planar and concave hillslopes, as the ones present in the study area, the partitioning of the flow provided by the use of the multiflow direction algorithm was consistent to the real situation (Seibert and McGlynn, 2007). Instead, this approach produced problematic flow paths if the flow of substances, such as sediments, was considered (Seibert and McGlynn, 2007). Thus, the use of the D-inf algorithm proposed by Tarboton (1997) was more accurate and less dispersive for a correct computation of the sediment connectivity in the IC parameter. For clarifying this concept, we then added this sentence: "... Multiflow direction algorithm distributed

the water flow to all neighboring downslope cells weighted according to slope angle, avoiding the flow concentration to particular lines sometimes unrealistic. In the case of planar and concave hillslopes, as the ones present in the study area, the partitioning of the flow provided by the use of the multiflow direction algorithm was consistent to the real situation (Seibert and McGlynn, 2007)...". References: Tarboton, D. G.: A new method for the determination of flow directions and upslope areas in grid digital elevation models, Water Resour. Res. 33, 2, 309– 319, doi:10.1029/96WR03137,1997. Seibert, J. and McGlynn, B. L.: A new triangular multiple flow direction algorithm for computing upslope areas from gridded digital elevation models, Water Resour. Res., 43, W04501, doi:10.1029/2006WR005128, 2007.

Comment 16 Line 15 p 6: why not considering a geodesic distance or a 3d distance, rather than a simply Euclidean distance? I'd assume that on a hilly slope, a 3d distance might be very different from a Euclidean one. Also, was this distance evaluated considering possible flow direction? I would assume that the possible direction/movement of a landslide would follow topography, and more specifically a shortest topographic travelling distance, rather than a simple 2d distance to the road network. Thus a 3d topographic distance might be more appropriate as a vulnerability index. Also in this paragraph, 'lowest distance' should be 'shortest distance.'

Response to Comment 16 We thank the reviewer for the suggestion. Flow direction distance between landslide and road represent a key variable for the analysis of such events. Flow direction-based distance is actually intrinsically included in the downslope component of IC. In the index of connectivity calculation, this distance is also weighted by the weighting factor W, which estimates the impedance to runoff and sediment fluxes due to properties of the local land use and soil surface (Cavalli et al., 2013; Crema and Cavalli, 2018). Thus, we do not consider a distance between landslide source area and road evaluated considering flow direction to avoid the presence of a parameter correlated with the index of connectivity reducing the redundancy and keeping independent the two features. Moreover, the shallow landslides triggered in the study area did not

follow established paths of the flow direction on the hillslopes where they occurred. In fact, these phenomena were not channeled, as in the case of typical debris flows or debris avalanches. For both these reasons, we decided to consider the distance between landslide source area and road as an Euclidean distance correspondent to the shortest trait between the landslide source area and a considered road sector. For a clarification of this concept, we also added the following sentence: "...Along with the DEM-derived predictor variables, the Euclidean distance from shallow landslide source area (DIST) was calculated, considering the shortest distance between the landslide source area and a considered road trait. The choice of an Euclidean distance was consistent to the types of slope failures present in the study area. The shallow landslides did not follow established paths of the flow direction on the hillslopes where they occurred. Moreover, they were not channeled, as in the case of typical debris flows or debris avalanches. Furthermore, a distance calculated along the flow direction was not considered to avoid redundancy with the parameter of sediment connectivity. In fact, sediment connectivity already took into account for the shortest paths along the flow direction in its downslope component (Cavalli et al., 2013; Crema and Cavalli, 2018)" References: Cavalli, M., Trevisani, S., Comiti, F. and Marchi, L.: Geomorphometric assessment of spatial sediment connectivity in small alpine catchments, Geomorphology 188, 31–41, doi:10.1016/j.geomorph.2012.05.007, 2013. Crema S., Cavalli M.: SedInConnect: A stand-alone, free and open source tool for the assessment of sediment connectivity, 25 Comp. Geosci., 111, 39-45, doi:10.1016/j.cageo.2017.10.009, 2018.

Comment 17 Line 15 p 9: is there a reference for this holdout bootstrap method?

Response to Comment 17 We indicated the most significant references for this holdout bootstrap method: Maindonald and Braun (2010): Maindonald, J. and Braun, W. J. (Eds.): Data analysis and graphics using R: an example based approach, Cambridge Series in Statistical and Probabilistic Mathematics, Cambridge, United Kingdom, 2010. McLachlan (1992): McLachlan, G. J. (Ed.): Discriminant analysis and statistical pattern recognition, John Wiley & Sons, New York, USA, 1992. Molinaro et al. (2005): Molinaro, A. M., Simon, R. and Pfeiffer, R. M.: Prediction error estimation: a comparison of resampling methods, Bioinf. 21, 3301-3307, doi:10.1093/bioinformatics/bti499, 2005.

Comment 18 Line 2 p 10: a buffer of 5 m from the middle of each road sector > what's the reasoning behind this buffer? Is this in line with the road size? Should it be varied considering main roads or minor roads?

Response to Comment 18 We thank the Referee for this comment. The chosen buffer of 5 m was consistent with the size of the roads present in the study area. These roads had similar sizes because they are all provincial or municipal routes with a width of the roadway between 3.5 and 5 m. In the case of other road typologies whose roadway widths are higher than 3.5-5 m (national roads, highways, roads with more than one lane for each direction of travel), it should be necessary increasing this buffer, to analyze completely the entire road trait. For a clarification of this concept, we added the following sentence: "...The chosen buffer of 5 m was consistent with the size of the roads present in the study area. These roads had similar sizes, with a width of the roadway ranging between 3.5 and 5 m...".

Comment 19 The first paragraph of the discussion is not needed. It is a repetition of the introduction.

Response to Comment 19 We thank the Referee for this suggestion. We removed this paragraph from the Discussions section.

Comment 20 Line 10 p 14. The authors state '' Instead, the proposed approach helps in filling the gaps and the limits still open in the definition of a reliable and, potentially, repeatable methodology.". However, I do not see how this was demonstrated. The method was replicated in their study case, so it is not that different from the previous literature they mentioned, where the methodologies were "developed and tested for particular geological/geomorphological settings."

Response to Comment 20 We thank the Referee for this suggestion and we rearranged

this concept. A reliable methodology for the classification of the susceptibility of different road traits was developed and tested. This methodology improved the definition of susceptibility thanks to: i) the implementation of a data-driven technique able to take into account also for the non-linear relationships between the predisposing factors and the response variable (road sector hit by shallow landslides); ii) the use of a parameter (the index of connectivity) that, if coupled with a landslide inventory, helps to assess the potential slope sediments mobilized by the landslide triggering which can reach the road network in downstream area, inserting also a proxy of landslide runout in the modeling of roads susceptibility.

Comment 21 Line 13 to 19> this is not about the current work. If anything, this should be mentioned when the authors justify the choice of the data-driven method, but it is not a result to discuss.

Response to Comment 21 We moved this part in the Introduction section, where we justified the choice of a data-driven model for the assessment of the roads susceptibility.

Comment 22 Line 7 to 15 p 15> this again is not about the current work. It should be eventually mentioned when the authors justify the choice of the IC, or to highlight similarities between their results and the mentioned works, which is not the case currently.

Response to Comment 22 We moved this part in the 3.1.1 section (Predictor variables), where we justified the choice of using the index of connectivity as a predictor variable for the definition of the roads susceptibility to shallow landslides.

Comment 23 Line 15 p 16 "They are in a buffer of less than 250 m, in particular between 50 and 200 m, respect to sectors hit in past, and they present morphological and connectivity features similar to threatened traits." > shouldn't the authors also include these locations (sectors hit in the past) in their assessment as reference data? If their model is meant to be feasible outside their study area and not the case-specific, it should be able to identify correctly all the elements, not only those triggered in one

specific event.

Response to Comment 23 We clarified this aspect. The model was tested using, as response variables, the road sectors hit by shallow landslides occurred in the study area during the three events recorded in last years whose inventories were available (27–28 April 2009, March/April 2013 and 28 February–2 March 2014 events). Thus, past hit sectors, mentioned in this part of the Discussions, referred to the road traits affected by shallow landslides triggered during the three known events. If other shallow landslides events causing damages to roads occurred in the study area during other rainfall events, we would use these data as a further validation of the reliability of the developed method.

Comment 24 The first paragraph of line 17 > 'Hence, more detailed scenarios of susceptibility changes about land use changes will take into account also for the morphological modifications linked to these changes, also using a higher resolution DEM (less than 1 m).' is this a future research line or a result?

Response to Comment 24 This aspect was not investigated in this paper and it could represent a future research line.

Please also note the supplement to this comment:
https://www.nat-hazards-earth-syst-sci-discuss.net/nhess-2017-457/nhess-2017-457-AC1-supplement.pdf
* * *
[Figure]

**Fig. 1.** Distribution of some predictor variables considering to model road susceptibility: a) SL; b) ASP; c) CURV; d) LEN; e) HEI; f) CA; g) CS; h) TWI; i) DIST.

**Supplement:**

**Table 1: Sensitivity of the different predictor variables on the accuracy for the training sets, the test sets and the final application of the model to the entire study area. The standard deviation of accuracy on training and test sets was of 0.01 for all the models, while the range of the 95 % confidence interval of AUC was of 0.02 for all the models.**

| GAM model | Mean accuracy of training sets (-) | Mean accuracy of test sets (-) | Mean AUC of the model (-) |
|---|---|---|---|
| 1 | 0.71 | 0.70 | 0.74 |
| 2 | 0.90 | 0.90 | 0.94 |
| 3 | 0.82 | 0.82 | 0.83 |
| 2 - (CURV, HEI, TWI) [threshold of selection equal to 90%] | 0.84 | 0.84 | 0.88 |
| 2 + (CA) [threshold of selection equalt to 50%] | 0.90 | 0.90 | 0.94 |
| 2 - (SL) | 0.74 | 0.74 | 0.78 |
| 2 - (CURV) | 0.87 | 0.87 | 0.92 |
| 2 - (HEI) | 0.85 | 0.77 | 0.91 |
| 2 - (CS) | 0.79 | 0.79 | 0.83 |
| 2 - (TWI) | 0.88 | 0.88 | 0.92 |
| 2 - (DIST) | 0.75 | 0.75 | 0.78 |
| 2 - (GEO) | 0.84 | 0.85 | 0.88 |
| 2 + (ASP) | 0.90 | 0.90 | 0.94 |
| 2 + (LEN) | 0.90 | 0.91 | 0.95 |

---

## Author Comment (AC2) · 3 Apr 2018

The authors are grateful to the Anonymous Referee #2, whose comments and suggestions will contribute towards an improvement of the final paper. We will also consider the suggestions proposed by the Anonymous Referee #1. Point-by-point replies to the Anonymous Referee #2's comments follow.

Comment 1 Dear Editor, The paper Estimation of the susceptibility of a road network to shallow landslides with the integration of the sediment connectivity" by Bordoni et al. present an interesting study case for landslide susceptibility applied to roads in a small area of north Apennines, Italy. One of the main contribution (as the authors

state in the title) is the integration of sediment connectivity in the statistical model, using different land use scenarios. From my point of view, the quality of the paper, the method used, the results and the discussion make the paper suitable for publication in NHESS journal. I suggest some highlights of this kind of studies in the introduction, discussions and conclusion parts of the paper. In my opinion, some minor issues must be solved before the publication.

Response to Comment 1 We thank the Referee for the appreciation of our work. In revising the paper, we considered all his suggestions and comments, which allowed to clarify several aspects and to improve the article overall quality.

Comment 2 (i) please provide more details in "the study area" section about the method used for land use mapping; also, you can move here some information about landslide inventory - data acquisition.

Response to Comment 2 We added several information about the method used for land use mapping. Regarding this aspect, we added this paragraph in "The study area" section of the paper: "...Land use maps of the study area have been available since 1954. Land use map of 1954 was realized by aerial photographs from Gruppo Aereo Italiano (Italian Aerial Group), with a resolution of 0.5 m. Further, the land use map of 1980 was obtained from photo interpretation at a scale of 1:50,000 from the TEM1 flight (scale 1:20.000). Land use maps of 2000, 2007, 2012 and 2015 were provided by the Lombardy Region and shared as part of the Infrastructure for Spatial Information in Lombardy (IIT) via the Geoportal (Lombardy Region Geoportal: http://www.cartografia.regione.lombardia.it/geoportale, last access: 11 December 2017). The map of 2000 was obtained from the photo interpretation of aerial images of Flight IT2000, with a resolution of 1 m. While, the land use map of 2007 was realized by using colour and infrared orthophotos from Flight IT2007, with a resolution of 0.5 m. The maps of 2012 and 2015, which corresponded to the actual situation, were realized through the photo-interpretation of aerial photos realized by Agency for Disbursement in Agriculture (AGEA). The photo-interpretation was also supported by auxiliary

data of Lombardy Region databases (e.g. Regional Agricultural Information System, Forest Types maps, map of the resident population, Archive of Integrated Activities production). The overall accuracies of maps obtained for Lombardy Region using this methodology was reported in Zaffaroni (2010) as approximately 95%. More detailed information about the method to realize these maps are available in Fasolini (2014)...". Furthermore, we moved the information about the inventory of affected road sectors in "The study area" section of the paper, adding this paragraph: "...A detailed inventory map of the road sectors affected by shallow landslides in the study area was prepared and used as response variable of the model. The inventory map of the affected road traits include all the sectors hit by the shallow landslides occurred in the study area during 27–28 April 2009, March/April 2013 and 28 February–2 March 2014 rainfall events. For 2009 event, color aerial photographs at a resolution of 15 cm acquired immediately after the event were examined (Persichillo et al., 2017). For 2013 event, affected road traits were identified by visual interpretation of Pleiades satellite images with a resolution of less than 1 m (Persichillo et al., 2017). For 2014 event, slope failures and affected roads immediately after the event were detected through field surveys; the identified phenomena were mapped through a GPS tool, whose resolution is less than 2.5 m...". References: Fasolini, D.: La cartografia dell'uso e copertura del suolo: uno strumento per rilevare il cambiamento del territorio lombardo, Regione Lombardia e ERSAF, 76–87, 2014. Persichillo, M. G., Bordoni, M. and Meisina, C.: The role of land use changes in the distribution of shallow landslides, Sci. Total Environ. 574, 924–937, doi:10.1016/j.scitotenv.2016.09.125, 2017. Zaffaroni, P.: Confronto fra CLC 2006 e DUSAF 2.1 della Regione Lombardia, ASITA, Brescia 9–12 November 2010, 2010.

Comment 3 (ii) the discussion part must rewrote, at least the first 4 paragraphs.

Response to Comment 3 We modified the first four paragraphs of the "Discussions" section. This modified part was provided: "...In this work, a methodology able to classify, in different susceptibility classes, the traits of a road network potentially hit by

sediments of landslides triggered above the road was developed and tested. Different Authors (Budetta, 2004; Hearn et al., 2008; Jaiswal et al., 2010a, 2010b, 2011; Quinn et al., 2010; Michoud et al., 2012; Tarolli et al., 2013; Bil et al., 2014, 2017; Penna et al., 2014; Ramesh and Anbazhagan, 2015; Tarolli and Dalla Sofia, 2016; Winter et al., 2016; Donnini et al., 2017; Pellicani et al., 2017; Postance et al., 2017; Martinovic et al., 2018) developed similar approaches in other geological/geomorphological settings, basing on the implementation of data-driven techniques for the estimation of road susceptibility. Data-driven models, that are based on the statistical relationships between predictors and response variables, depend strictly on the reliability of the inventories of the response variable (Guzzetti et al., 2006; Corominas et al., 2014). Besides this limitation, data-driven are most flexible to be used at different scales of analysis (from site-specific to regional scale) and do not require a lot of data not easily to be estimated as for the physically-based models (Corominas et al., 2014). For the first time, the proposed methodology allowed to implement a data-driven technique (GAM method) able to take into account also for the non-linear relationships between the predictors and the response variable (road sector hit by shallow landslides). In particular, the models some of the input predictors as non-linear variables (in this case, slope curvature, catchment slope, topographic wetness index, distance from shallow landslides source area, index of connectivity), understanding better the complex relationships which are present in an area between predisposing factors and susceptible roads (Philips, 2006; Goetz et al., 2011). Moreover, before building the model, the individuation of the most important predictor variables among the generally used predisposing factors leads to improve the knowledge about mechanisms which regulate the location of the damaged roads in such an area, avoiding for collinearity and bias that could reduce the reliability of the susceptibility estimation (Farrar and Glauber, 1967; Hosmer and Lemeshow, 1990; Bai et al., 2010). The robustness of the proposed methodology was also confirmed by the low confidence degree measured for the created models (Petschko et al., 2014). The first reconstructed susceptibility model (Model 1) takes into account for the most important predisposing factors in the study area, chosen among those morphological, hydrological and geological parameters taken into account for these analyses in different contexts by other Authors (Budetta, 2004; Jaiswal et al., 2010a, 2010b, 2011; Quinn et al., 2010; Michoud et al., 2012; Bil et al., 2014, 2017; Penna et al., 2014; Ramesh and Anbazhagan, 2015; Pellicani et al., 2017). The reliability of the model is quite fair, as testified by its AUC value (0.73) and by its high value of FP and TN indexes (22.3 and 45.8%, respectively). According to Model, most susceptible road segments are those located downstream to slopes characterized by high slope gradient (> 20°), limited height (< 50 m) and with shallow landslides triggering zones located very close to the road network (40-100 m). These settings are very widespread in the entire study area (Bordoni et al., 2015; Persichillo et al., 2016, 2018), but these particular features are not enough to discriminate more accurately those routes where damages provoked by sediments mobilized by shallow landslides are probable...". In these paragraph, following references will be added to the revised version of the paper: Bil et al. (2017): Bil, M., Andrasik, R., Kubecek, J., Krivankova, Z. and Vodak, R.: RUPOK: An online landslide risk tool for road networks, in: Advancing culture of living with landslides, edited by: Mikos, M., Vilimek, V., Yin, Y., and Sassa, K., Springer, Cham, 19-26, 2017. Donnini et al. (2017): Donnini, M., Napolitano, E., Salvati, P., Ardizzone, F., Bucci, F., Fiorucci, F., Santangelo, M., Cardinali, M. and Guzzetti F.: Impact of event landslides on road networks: a statistical analysis of two Italian case studies, Landslides 14, 4, 1521–1535, doi:10.1007/s10346-017-0829-4, 2017. Hearn et al. (2008): Hearn, G., Hunt, T., Aubert, J. and Howell, J.: Landslide impacts on the road network of Lao PDR and the feasibility of implementing a slope management programme, South East Asia Community Access Programme (SEACAP), Department for International Development, United Kingdom. Martinovic et al. (2018): Martinovic, K., Gavin, K., Reale, C. and Mangan, C.: Rainfall thresholds as a landslide indicator for engineered slopes on the Irish Rail network, Geomorphology 306, 40-50, doi:10.1016/j.geomorph.2018.01.006, 2018. Penna et al. (2014): Penna, D., Borga, M., Aronica, G. T., Brigandì, G. and Tarolli, P.: The influence of grid resolution on the prediction of natural and road-related shallow landslides. Hydrol. Earth Syst. Sci. 18, 6, 2127–2139, doi:10.5194/hess-18-2127-2014,

2014. Postance et al. (2017): Postance, B., Hillier, J., Dijkstra, T. and Dixon, N.: Extending natural hazard impacts: an assessment of landslide disruptions on a national road transportation network, Environ. Res. Let. 12, 1, 14010, doi:10.1088/1748-9326/aa5555, 2017. Tarolli et al. (2013): Tarolli, P., Calligaro, S., Cazorzi, F. and Dalla Fontana, G.: Recognition of surface flow processes influenced by roads and trails in mountain areas using high-resolution topography, Eur. J. Remote Sens. 46, 176–197, doi:10.5721/EuJRS20134610, 2013. Tarolli and Sofia (2016): Tarolli, P. and Sofia, G.: Human topographic signatures and derived geomorphic processes across landscapes, Geomorphology 255, 140–161, doi:10.1016/j.geomorph.2015.12.007, 2016. Winter et al. (2016): Winter, M. G., Shearer, B., Palmer, D., Peeling, D., Harmer, C. and Sharpe J.: The economic impact of landslides and floods on the road network, Procedia Eng. 143, 1425-1434, doi:10.1016/j.proeng.2016.06.168, 2016. Other references present in this correction are already inserted in the manuscript.

Comment 4 (iii) please change the Fig. 9, in order to make it representative according to the associated legend;

Response to Comment 4 We modified this figure for improving its comprehension. Modified Figure 9 was attached to these responses.

Comment 5 and (iv) try to find other color for the roads (at least in the Figures 1, 3, and 9) for increasing the contrast and readability of the map.

Response to Comment 5 We modified the colors of road network in Figures 1 and 3 to improve its visualization. Modified Figures 1 and 3 were attached to these responses.

Comment 6 Other minor problems you will find in the *.pdf file attached..

Response to Comment 6 We thank the Referee also for these suggestions. We considered all these minor revisions proposed by the Referee and we will insert them in the modified version of the manuscript.
* * *
2017-457, 2018.

[Figure]

**Fig. 1.** Modified Figure 1 of the manuscript

**Fig. 2.** Modified Figure 3 of the manuscript

a)

b)

c)

95% bootstrap confidence intervals

| | | | |
|---|---|---|---|
| ■ 0.0 - 0.1 | ■ 0.1 - 0.2 | ■ 0.2 - 0.3 | ■ 0.3 - 0.4 |
| ■ 0.4 - 0.5 | ■ 0.5 - 0.6 | ■ 0.6 - 0.7 | ■ 0.7 - 0.8 |
| ■ 0.8 - 0.9 | ■ 0.9 - 1.0 | | |
| ▢ Study area | | | |

**Fig. 3.** Modified Figure 9 of the manuscript

---

## Author Comment (AC3) · 17 Apr 2018

The authors are grateful to the Anonymous Referees, whose comments and suggestions will contribute towards an improvement of the final paper. In the revised version of the manuscript ,attached to this comment, modified and added parts are highlighted in yellow. Point-by-point replies to the Anonymous Referees' comments follow.

Response to Referee #1 Comment 1 The work is an interesting contribution to the journal, and it provides new insights on the relationship between roads and landslides, from a land management point of view. I have, however, some major concerns that should be addressed before the paper is ready for publication.

[Figure]

Response to Comment 1 We thank the Referee for the appreciation of our work. In revising the paper, we considered all the suggestions and comments, which allowed to clarify several aspects and to improve the article overall quality.

Comment 2 According to the manuscript, the variables of the model were selected using the AIC criterion, but the authors do not explain which one: the backward approach, the forward one or the backward-forward? In the Forward method, one starts with an empty model, and iterate over all features. For each feature, the model is trained, and one select the feature which yields the best model according to a specific metric. Similarly, further features that yield the best improvement when combined with the already selected ones are added. In the backward method we start with all features, and iteratively remove that one whose removal least hurt the performance, or leads to the biggest improvement. Therefore, the models selected by forward selection or backwards elimination might not be the same, even using the same model selection criterion. Also, the authors do not specify what criterion is considered to define the 'best' model achieved when adding/removing a feature. They only speak about the final performance of the model but do not provide any comparison between results obtained by adding or removing variables.

Response to Comment 2 We clarified this aspect about the implementation of the GAM model for road susceptibility estimation. For the selection of the explanatory variables, we used the 'step.Gam' command of the R package 'gam'. This command, as explained in the library manual (available at URL https://cran.r-project.org/web/packages/gam/gam.pdf), allows to choose any of the three approaches. We choose to select the variables allowing both directions in the step-wise search, using the option direction="both" in issuing the step.Gam command. The selected "best" model is the one that minimizes the Akaike Iteration Criterion statistic. This procedure was repeated 100 times using 100 bootstrap extractions form the same dataset. The final model was chosen according to an acceptance threshold of 80%. In the response to comment 3, it is present a reasonable justification of this threshold. This clarification was added at pag. 10 lines 3-8 of the revised version of the manuscript.

Comment 3 A further question arises: what's the reason behind choosing 80% as the threshold for variable acceptance? There is no justification for this choice, aside from an author preference. While I do understand that 80% is a high number, what is the difference in the quality of the results at the change of this threshold? The authors should consider this a bit more in detail [i.e. as for the previous point, does removing/adding one variable or the other improve the results significantly? What if we select variables chosen more than 50% or 90% of the times?]. Addressing these two points would also improve the discussion in Chapt .4.2.1.

Response to Comment 3 As suggested by the Referee in his Comment 2, we performed a sensitivity analysis to assess the role of each predictor variable on the accuracy of the GAM models. This analysis allowed also evaluating the change in predictive accuracy related to adding or removing a set of predictors according to a threshold of selection different than the used 80% or related to adding or removing a particular predictor. It is important to highlight that the results of this sensitivity analysis shown in the paper referred to the susceptibility model which had the best predictive accuracy, that is Model 2 considering all the predictors selected using the threshold of 80% and the index of connectivity calculated in a linear way. Instead, the quantitative changes on the predictive accuracy related to different sets of predictors were similar also considering Model 1 (all the predictors selected using the threshold of 80% without considering the index of connectivity) and Model 3 (all the predictors selected using the threshold of 80% and the index of connectivity calculated in the non-linear way). Table 4 added to the revised version of the manuscript showed the results of this sensitivity analysis. First, the effects on the model accuracy related to a change of the value of selection threshold used for choosing the predictor variables used for the creation of the final susceptibility model were evaluated. According to the percentages of selection of each variable in the 100-fold bootstrap procedure (Tab. 2 of the manuscript), also thresholds

of 50% and 90% of selection frequency were considered and compared to the used threshold of 80%. A threshold of selection frequency lower than 50% was not considered significant. Considering a threshold equal to 50%, also CA (chosen as a linear variable) had to be inserted for modeling the susceptibility. Instead, the mean predictive accuracy of the model, estimated in terms of AUC values, did not change, for both the training sets, the test sets and the final model. The difference in the predictive accuracy was lower than 0.01. Instead, concerning a threshold equal to 90%, CURV, HEI and TWI had to be removed, because their frequency selections were between 85 and 88%. In this case, the mean predictive accuracy of the best model (Model 2) decreased from 0.90 to 0.84 and from 0.94 to 0.88 for training/test sets and for the final models, respectively. Removing a predictor or a set of these from the susceptibility model caused a decrease of the accuracy due to a reduction in explaining the physical relations between the predisposing factors and the resulting effects on the response variable, in this case represented by the road sectors hit by shallow landslides. These results demonstrated that a threshold of selection of the predictors equal to 80% allowed to obtain the sets of predisposing factors able to estimate in the best reliable and effective way the susceptibility of the road network to be affected by shallow landslides. Furthermore, a sensitivity analysis of the different predictors considered as predisposing factors for road susceptibility was performed. This analysis consisted in running one of the models created considering a threshold of selection frequency equal to 80% (Model 1, 2, 3), removing each time one of the selected predictors or adding each time one of the other predictors, whose frequency of selection was lower than 80%. In this way, the sensitivity of the model to each predictor could be quantified. Starting from Model 2, which was the best in terms of reliability, the removal of a particular predictor could affect the accuracy. Removing SL or DIST caused a reduction of the predictive accuracy, for both training sets, test sets and final models, of 0.15-0.16. Instead, this reduction was lower than the one quantified if in the model IC was not taken into account (Model 1). In fact, the absence of IC provoked a decrease in the accuracy of 0.19-0.20. The removal of CS caused a moderate reduction of the accuracy, correspondent to 0.11. While, removing one of the other chosen parameters (CURV, HEI, TWI, GEO) provoked only a slight decrease in the predictive accuracy, in the order of 0.02-0.06. Moreover, adding alternatively to the chosen predictors one of the other predisposing factors (CA, ASP, LEN) did not modify significantly the reliability of the models. The predictive accuracy improved at most 0.01 for both training sets, test sets and final models. These results confirmed the significant sensitivity of the susceptibility model to IC, especially the one estimated in a linear way. Neglecting IC in these models caused a big decrease in the effectiveness, which affects significantly the susceptibility classification of the road network. Furthermore, SL and DIST also affected significantly the accuracy of the final susceptibility model and had to be considered for obtaining a correct classification of road network. The models were more slightly sensitive to the other chosen predictors (CURV, HEI, TWI, GEO). Instead, the leakage of only one of those parameters could decrease the final reliability of the road susceptibility. The models were not sensitive to the other considered predisposing factors (CA, ASP, LEN). Thus, these parameters did not allow for a further improvement of the susceptibility models reliability and could be correctly excluded from the models. These results confirmed further the goodness of choosing a threshold of selection frequency of the predictors equal to 80%. It is important to note that the standard deviation of accuracy on training and test sets was of 0.01 for all the models, while the range of the 95 % confidence interval of AUC was of 0.02 for all the models. We added information about this analysis at pag. 16 lines 9-34 and pag. 17 lines 1-23 of the revised version of the manuscript.

Comment 4 Another point is that currently there are no rational formulations for the indices that are kept or removed, other than the fact that they are a mathematical construct. What I mean is: is there a physical meaning behind the rejection or acceptance of such parameters? The description of the IC for the area helps to interpret its importance in the model, and the reason behind the increased quality of models that do include it in one way or another. However, the authors should also describe the other indices about the road network in their study (not just as a general statement on

why they are important, as done in Chapt. 3.1.1), to justify their choice or confirm the model assumption. This would also help 'balance' the paper more: as of now, the focus on connectivity seems unbalanced, and similar to the previous work by (Persichillo, Bordoni, Cavalli, Crema, & Meisina, 2018).

Response to Comment 4 As done in the original version of the manuscript for IC, we better analyzed the distribution of the other predictor variables considered for building the susceptibility models of the study area. This allowed integrating the analysis of the role played by IC in road susceptibility, giving indications also on the role played by the other features. The maps of the distribution of some predictors (SL, ASP, CURV, LEN, HEI, CA, CS, TWI, DIST) were present in Fig. 1 attached to these responses. The distribution of the bedrock geological formations (GEO) in the study area was already shown in Fig. 1 of the manuscript. The procedure adopted for the selection of the most significant variables used for the creation of the susceptibility models excluded ASP, LEN and CA parameters. Distribution of ASP in the study area (Fig. 1b) showed that the exposition of the slopes close to the roads was very variable, without the identification of peculiar features. While, LEN (Fig. 1d) and CA (Fig. 1f) values close to the road sectors were in a quite narrow range, between 2 and 150 m and around 102 m2, respectively. The particular distributions of these parameters confirmed their not significant roles in the evaluation of the road susceptibility. Thus, they could be correctly not considered in GAM models. Concerning the predictors selected by the 100-fold bootstrap procedure, roads were located especially close to hillslopes of medium-high SL (higher than 10°, except for the routes located in the floors of the river valleys; Fig. 1a), limited HEI (lower than 50 m; Fig. 1e) and with shallow landslides triggering zones located very close to the road network (lower than 150 m; Fig. 1i). In fact, the affected road sectors were generally road segments downstream to slopes characterized by high slope gradient (> 20°), limited height (< 50 m) and with shallow landslides triggering zones located very close to the road network (40-100 m). These sectors were correctly classified as susceptible by the best model (Model 2). Also CS had an important effect on the susceptibility of a road to be hit by shallow failures. Roads were located close to hillslopes with very low (0-5°) or very high (20-31°) CS values (Fig. 1g). Affected roads, classified as susceptible by the implemented models, corresponded to road traits located in correspondence of very high values of CS, generally between 20-28°. As highlighted in the Response to Comment 3, CURV, TWI and GEO were selected by the methodology as significant predictors, but they had a lower effect on the accuracy of the models. This meant that they explained less than the other selected predictors the susceptibility of a road to shallow landslides. Instead, CURV values (Fig. 1c) close to the roads were generally slightly negative (lower than -0.05) and the affected sectors were in correspondence of the lowest CURV values (around -0.40). TWI (Fig. 1h) was generally positive in correspondence of road traits, with values higher than 5 close to sectors affected by shallow landslides. Moreover, damaged road traits were mainly located in areas where GEO was composed of medium low-permeable arenaceous conglomeratic materials (Monte Arzolo Sandstones, Rocca Ticozzi Conglomerates) or impermeable silty-sandy marly bedrock (Montù Beccaria Formation, Sant'Agata Fossili Marls). We added explanation about this analysis at pag. 15 lines 26-30, pag. 16 lines 9-34 and pag. 17 lines 1-23 of the revised version of the manuscript.

Comment 5 Some minor comments arose as I read the manuscript. English needs polishing. Some parts are too 'colloquial' (e.g. '' It is also worth noting that") or have some English mistakes, mostly in the first part of the manuscript e.g. "the evaluation of the importance of considering or neglecting sediment connectivity" is redundant, you can simply state 'the importance of considering sediment connectivity'.

Response to Comment 5 We thank the Referee for this suggestion. We carefully performed a detailed revision of English to clarify several unclear sections, to delete colloquial sentences and mistakes and to improve the overall quality of the manuscript. We also considered the suggested correction of the Referee, changing the sentence "...the evaluation of the importance of considering or neglecting sediment connectivity..." in "...the importance of considering connectivity...", at pag. 3 line 33 of the revised version

of the manuscript.

Comment 6 Line 31 p 3: "in the routes distribution that could be affected by shallow landslides" > is the distribution affected by shallow landslides or are the roads affected by it?

Response to Comment 6 We rearranged this sentence to clarify the expressed concept. We meant that the road sectors of a particular area, prone to shallow landslides, could be affected by phenomena triggered in the closest slopes, which could damage the roads themselves. Thus, we modified this sentence in: "...the road sectors potentially affected by shallow landslides...". This modification was added at pag. 4 line 2 of the revised version of the manuscript.

Comment 7 Line 29 p 4. "The road sectors were built in correspondence of the valley floors or hillside, cutting a portion of a hillslope in correspondence of its medium part realising a halfway road" > this sentence is not clear, what Is a halfway road? Medium part of what, of the hillslope?

Response to Comment 7 We rearranged this sentence to clarify the expressed concept. In the study area, the roads were built in correspondence of the valley floors or in the medium part of a hillslope, cutting its continuity. This sentence was then modified in: "...The roads were built in correspondence of the valley floors or in the medium part of a hillslope, cutting its continuity...". This sentence was added at pag. 5 lines 10-11 of the revised version of the manuscript.

Comment 8 Line 6 p 5: '30% of these shallow landslides WERE triggered in vineyards

Response to Comment 8 We added "were" where it leaked, to correct this grammatical error, at pag. 5 line 20 of the revised version of the manuscript.

Comment 9 Line 10 p 5. What is a b2 type?

Response to Comment 9 In the set of the shallow landslides hitting roads of the study area, 24 failures (5% of the total number) were roto-translational slides affecting the

trench of a cut realized for building the road itself. Zizioli et al. (2013) and Persichillo et al. (2018) named these failures as "B2" type, thus we decided to recall this type of landslides with the same term. This clarification was inserted at pag. 5 lines 23-25 of the revised version of the manuscript. References are present in the References section of the revised version of the manuscript.

Comment 10 Line 19 p 9 'to discriminate affected or not road sectors' this is redundant. It is clear that by discriminating affected road sections, it would remove those not affected.

Response to Comment 10 We thank the Referee for this suggestion. We modified this sentence in: "...to discriminate affected road sectors...". This was modified at pag. 10 line 29 of the revised version of the manuscript.

Comment 11 Abstract needs rewording. Some concept are introduced without the reader knowing what they are, e.g. 'The random partition of the dataset used for building the model in two parts (training and test subsets), within a 100-fold bootstrap procedure.'

Response to Comment 11 We thank the Referee for this revision. We modified the Abstract, in those unclear parts to introduce better several concepts not completely presented before. In particular, the modified parts were at pag. 1 lines 14-32.

Comment 12 Literature in the introduction could be improved, e.g. about road networks and landslides (Bíl, Andrásik, Kubecek, Krivánková, & Vodák, 2017; Donnini et al., 2017; Hearn, Hunt, Aubert, & Howell, 2008; Martinovi'c, Gavin, Reale, & Mangan, 2018; Penna, Borga, Aronica, Brigandì, & Tarolli, 2014; Postance, Hillier, Dijkstra, & Dixon, 2017; P. Tarolli, Calligaro, Cazorzi, & Dalla Fontana, 2013; Paolo Tarolli & Sofia, 2016) and about road-landslides and climate changes (Klose, Auerbach, Herrmann, Kumerics, & Gratzki, 2017; Michaelides, 2014; Strauch, Raymond, Rochefort, Hamlet, & Lauver, 2015).

[Figure]

Response to Comment 12 We improved the literature review about the interactions between landslides and roads and about the effects of climate change in events of landslides triggering affecting road networks. In particular, we added in the manuscript the following references related to the interactions between landslides and roads: Bil et al. (2017): Bil, M., Andrasik, R., Kubecek, J., Krivankova, Z. and Vodak, R.: RUPOK: An online landslide risk tool for road networks, in: Advancing culture of living with landslides, edited by: Mikos, M., Vilimek, V., Yin, Y., and Sassa, K., Springer, Cham, 19-26, 2017. Donnini et al. (2017): Donnini, M., Napolitano, E., Salvati, P., Ardizzone, F., Bucci, F., Fiorucci, F., Santangelo, M., Cardinali, M. and Guzzetti F.: Impact of event landslides on road networks: a statistical analysis of two Italian case studies, Landslides 14, 4, 1521–1535, doi:10.1007/s10346-017-0829-4, 2017. Hearn et al. (2008): Hearn, G., Hunt, T., Aubert, J. and Howell, J.: Landslide impacts on the road network of Lao PDR and the feasibility of implementing a slope management programme, South East Asia Community Access Programme (SEACAP), Department for International Development, United Kingdom. Martinovic et al. (2018): Martinovic, K., Gavin, K., Reale, C. and Mangan, C.: Rainfall thresholds as a landslide indicator for engineered slopes on the Irish Rail network, Geomorphology 306, 40-50, doi:10.1016/j.geomorph.2018.01.006, 2018. Penna et al. (2014): Penna, D., Borga, M., Aronica, G. T., Brigandì, G. and Tarolli, P.: The influence of grid resolution on the prediction of natural and road-related shallow landslides. Hydrol. Earth Syst. Sci. 18, 6, 2127–2139, doi:10.5194/hess-18-2127-2014, 2014. Postance et al. (2017): Postance, B., Hillier, J., Dijkstra, T. and Dixon, N.: Extending natural hazard impacts: an assessment of landslide disruptions on a national road transportation network, Environ. Res. Let. 12, 1, 14010, doi:10.1088/1748-9326/aa5555, 2017. Tarolli et al. (2013): Tarolli, P., Calligaro, S., Cazorzi, F. and Dalla Fontana, G.: Recognition of surface flow processes influenced by roads and trails in mountain areas using high-resolution topography, Eur. J. Remote Sens. 46, 176–197, doi:10.5721/EuJRS20134610, 2013. Tarolli and Sofia (2016): Tarolli, P. and Sofia, G.: Human topographic signatures and derived geomorphic processes across landscapes, Geomorphology 255, 140–

161, doi:10.1016/j.geomorph.2015.12.007, 2016. Winter et al. (2016): Winter, M. G., Shearer, B., Palmer, D., Peeling, D., Harmer, C. and Sharpe J.: The economic impact of landslides and floods on the road network, Procedia Eng. 143, 1425-1434, doi:10.1016/j.proeng.2016.06.168, 2016. Moreover, we added the following references related to the effects of climate change in triggering events of landslides affecting road networks: Klose et al. (2017): Klose, M., Auerbach, M., Herrmann, C., Kumerics, C. and Gratzki, A.: Landslide hazards and climate change adaptation of transport infrastructures in Germany, in: Advancing culture of living with landslides, edited by: Mikos, M., Vilimek, V., Yin, Y., and Sassa, K., Springer, Cham, 535-541, 2017. Michaelides (2014): Michaelides, S.: Vulnerability of transportation to extreme weather and climate change, Nat. Hazards 72, 1, 1–4, doi:10.1007/s11069-013-0975-5, 2014. Strauch, R. L., Raymond, C. L., Rochefort, R. M., Hamlet, A. F. and Lauver, C.: Adapting transportation to climate change on federal lands in Washington State, U.S.A, Clim. Change 130, 2, 185–199, doi:10.1007/s10584-015-1357-7, 2015.

Comment 13 Line 30 p 5: slope aspect (ASP), slope curvature (CURV) > these would be better defined as simply aspect and curvature.

Response to Comment 13 We thank the Referee for this suggestion. We defined these two geomorphological parameters only as "aspect" and "curvature", respectively.

Comment 14 Also, what is the 'slope height'?

Response to Comment 14 Slope height represents the elevation difference between the source area of a shallow landslide and the bottom of the hillslope where this failure occurred. For clarifying this concept, we added the following sentence at pag. 7 lines 1-2 of the revised version of the manuscript: "...Slope height (HEI) represented the elevation difference between the source area of a shallow landslide and the bottom of the hillslope where this failure occurred".

Comment 15 Line 10 p 6. Why using the multiflow algorithm? Wouldn't it be more consistent to use the D-Infinity since the sediment connectivity is also computed through

D-inf, which is more accurate and less dispersive, especially on hillslopes?

Response to Comment 15 Multiflow direction algorithm was used for the computation of the catchment area and the catchment slope. Catchment area and catchment slope were used as proxies for soil moisture and soil depth and for the destabilizing forces upstream that can provoke the development of a landslide, respectively. Multiflow direction algorithm distributed the water flow to all neighboring downslope cells weighted according to slope angle, avoiding the flow concentration to particular lines sometimes unrealistic. In the case of planar and concave hillslopes, as the ones present in the study area, the partitioning of the flow provided by the use of the multiflow direction algorithm was consistent to the real situation (Seibert and McGlynn, 2007). Instead, this approach produced problematic flow paths if the flow of substances, such as sediments, was considered (Seibert and McGlynn, 2007). Thus, the use of the D-inf algorithm proposed by Tarboton (1997; Tarboton, D. G.: A new method for the determination of flow directions and upslope areas in grid digital elevation models, Water Resour. Res. 33, 2, 309– 319, doi:10.1029/96WR03137, 1997) was more accurate and less dispersive for a correct computation of the sediment connectivity in the IC parameter. For clarifying this concept, we then added this sentence: "...Multiflow direction algorithm distributed the water flow to all neighboring downslope cells weighted according to slope angle, avoiding the flow concentration to particular lines sometimes unrealistic. In the case of planar and concave hillslopes, as the ones present in the study area, the partitioning of the flow provided by the use of the multiflow direction algorithm was consistent to the real situation (Seibert and McGlynn, 2007)...". This clarification was inserted at pag. 7 lines 3-7 of the revised version of the manuscript. References are present in the References section of the revised version of the manuscript.

Comment 16 Line 15 p 6: why not considering a geodesic distance or a 3d distance, rather than a simply Euclidean distance? I'd assume that on a hilly slope, a 3d distance might be very different from a Euclidean one. Also, was this distance evaluated considering possible flow direction? I would assume that the possible direction/movement

of a landslide would follow topography, and more specifically a shortest topographic travelling distance, rather than a simple 2d distance to the road network. Thus a 3d topographic distance might be more appropriate as a vulnerability index. Also in this paragraph, 'lowest distance' should be 'shortest distance.'

Response to Comment 16 We thank the reviewer for the suggestion. Flow direction distance between landslide and road represent a key variable for the analysis of such events. Flow direction-based distance is actually intrinsically included in the downslope component of IC. In the index of connectivity calculation, this distance is also weighted by the weighting factor W, which estimates the impedance to runoff and sediment fluxes due to properties of the local land use and soil surface (Cavalli et al., 2013; Crema and Cavalli, 2018). Thus, we do not consider a distance between landslide source area and road evaluated considering flow direction to avoid the presence of a parameter correlated with the index of connectivity reducing the redundancy and keeping independent the two features. Moreover, the shallow landslides triggered in the study area did not follow established paths of the flow direction on the hillslopes where they occurred. In fact, these phenomena were not channeled, as in the case of typical debris flows or debris avalanches. For both these reasons, we decided to consider the distance between landslide source area and road as an Euclidean distance correspondent to the shortest trait between the landslide source area and a considered road sector. For a clarification of this concept, we also added the following sentence at pag. 7 lines 12-18 of the revised version of the manuscript: "... Along with the DEM-derived predictor variables, the Euclidean distance from shallow landslide source area (DIST) was calculated, considering the shortest distance between the landslide source area and a considered road trait. The choice of an Euclidean distance was consistent to the types of slope failures present in the study area. The shallow landslides did not follow established paths of the flow direction on the hillslopes where they occurred. Moreover, they were not channeled, as in the case of typical debris flows. Furthermore, a distance calculated along the flow direction was not considered to avoid redundancy with the parameter of sediment connectivity. In fact, sediment connectivity already took

into account for the shortest paths along the flow direction in its downslope component (Cavalli et al., 2013; Crema and Cavalli, 2018)...". References are present in the References section of the revised version of the manuscript.

Comment 17 Line 15 p 9: is there a reference for this holdout bootstrap method?

Response to Comment 17 We indicated the most significant references for this holdout bootstrap method, at pag. 10 line 24 of the revised version of the manuscript: Maindonald and Braun (2010): Maindonald, J. and Braun, W. J. (Eds.): Data analysis and graphics using R: an example based approach, Cambridge Series in Statistical and Probabilistic Mathematics, Cambridge, United Kingdom, 2010. McLachlan (1992): McLachlan, G. J. (Ed.): Discriminant analysis and statistical pattern recognition, John Wiley & Sons, New York, USA, 1992. Molinaro et al. (2005): Molinaro, A. M., Simon, R. and Pfeiffer, R. M.: Prediction error estimation: a comparison of resampling methods, Bioinf. 21, 3301-3307, doi:10.1093/bioinformatics/bti499, 2005.

Comment 18 Line 2 p 10: a buffer of 5 m from the middle of each road sector > what's the reasoning behind this buffer? Is this in line with the road size? Should it be varied considering main roads or minor roads?

Response to Comment 18 We thank the Referee for this comment. The chosen buffer of 5 m was consistent with the size of the roads present in the study area. These roads had similar sizes because they are all provincial or municipal routes with a width of the roadway between 3.5 and 5 m. In the case of other road typologies whose roadway widths are higher than 3.5-5 m (national roads, highways, roads with more than one lane for each direction of travel), it should be necessary increasing this buffer, to analyze completely the entire road trait. For a clarification of this concept, we added the following sentence: "...The chosen buffer of 5 m was consistent with the size of the roads present in the study area. These roads had similar sizes, with a width of the roadway ranging between 3.5 and 5 m...". This was added at pag. 11 lines 10-12 of the revised version of the manuscript.

[Figure]

Comment 19 The first paragraph of the discussion is not needed. It is a repetition of the introduction.

Response to Comment 19 We thank the Referee for this suggestion. We removed this paragraph from the Discussions section.

Comment 20 Line 10 p 14. The authors state '' Instead, the proposed approach helps in filling the gaps and the limits still open in the definition of a reliable and, potentially, repeatable methodology.''. However, I do not see how this was demonstrated. The method was replicated in their study case, so it is not that different from the previous literature they mentioned, where the methodologies were "developed and tested for particular geological/geomorphological settings."

Response to Comment 20 We thank the Referee for this suggestion and we rearranged this concept. A reliable methodology for the classification of the susceptibility of different road traits was developed and tested. This methodology improved the definition of susceptibility thanks to: i) the implementation of a data-driven technique able to take into account also for the non-linear relationships between the predisposing factors and the response variable (road sector hit by shallow landslides); ii) the use of a parameter (the index of connectivity) that, if coupled with a landslide inventory, helps to assess the potential slope sediments mobilized by the landslide triggering which can reach the road network in downstream area, inserting also a proxy of landslide runout in the modeling of roads susceptibility. This clarification was added at pag. 15 lines 2-30 of the revised version of the manuscript.

Comment 21 Line 13 to 19> this is not about the current work. If anything, this should be mentioned when the authors justify the choice of the data-driven method, but it is not a result to discuss.

Response to Comment 21 We moved this part in the Introduction section, at pag. 3 lines 4-7 of the revised version of the manuscript, where we justified the choice of a data-driven model for the assessment of the roads susceptibility.

[Figure]

Comment 22 Line 7 to 15 p 15> this again is not about the current work. It should be eventually mentioned when the authors justify the choice of the IC, or to highlight similarities between their results and the mentioned works, which is not the case currently.

Response to Comment 22 We moved this part in the 3.1.1 section (Predictor variables), at pag. 7 lines 30-34 and pag. 8 lines 1-5 of the revised version of the manuscript, where we justified the choice of using the index of connectivity as a predictor variable for the definition of the roads susceptibility to shallow landslides.

Comment 23 Line 15 p 16 "They are in a buffer of less than 250 m, in particular between 50 and 200 m, respect to sectors hit in past, and they present morphological and connectivity features similar to threatened traits." > shouldn't the authors also include these locations (sectors hit in the past) in their assessment as reference data? If their model is meant to be feasible outside their study area and not the case-specific, it should be able to identify correctly all the elements, not only those triggered in one specific event.

Response to Comment 23 The model was tested using, as response variables, the road sectors hit by shallow landslides occurred in the study area during the three events recorded in last years whose inventories were available (27–28 April 2009, March/April 2013 and 28 February–2 March 2014 events). Thus, past hit sectors, mentioned in this part of the Discussions, referred to the road traits affected by shallow landslides triggered during the three known events. If other shallow landslides events causing damages to roads occurred in the study area during other rainfall events, we would use these data as a further validation of the reliability of the developed method.

Comment 24 The first paragraph of line 17 > 'Hence, more detailed scenarios of susceptibility changes about land use changes will take into account also for the morphological modifications linked to these changes, also using a higher resolution DEM (less than 1 m).' is this a future research line or a result?

Response to Comment 24 This aspect was not investigated in this paper and it could

represent a future research line.

Response to Referee #2 Comment 1 Dear Editor, The paper Estimation of the susceptibility of a road network to shallow landslides with the integration of the sediment connectivity" by Bordoni et al. present an interesting study case for landslide susceptibility applied to roads in a small area of north Apennines, Italy. One of the main contribution (as the authors state in the title) is the integration of sediment connectivity in the statistical model, using different land use scenarios. From my point of view, the quality of the paper, the method used, the results and the discussion make the paper suitable for publication in NHESS journal. I suggest some highlights of this kind of studies in the introduction, discussions and conclusion parts of the paper. In my opinion, some minor issues must be solved before the publication.

Response to Comment 1 We thank the Referee for the appreciation of our work. In revising the paper, we considered all his suggestions and comments, which allowed to clarify several aspects and to improve the article overall quality.

Comment 2 (i) please provide more details in "the study area" section about the method used for land use mapping; also, you can move here some information about landslide inventory - data acquisition.

Response to Comment 2 We added several information about the method used for land use mapping. Regarding this aspect, we added this paragraph in "The study area" section of the paper, at pag. 4 lines 14-26 of the revised version of the manuscript: "...Land use maps of the study area have been available since 1954. Land use map of 1954 was realized by aerial photographs from Gruppo Aereo Italiano (Italian Aerial Group), with a resolution of 0.5 m. Further, the land use map of 1980 was obtained from photo interpretation at a scale of 1:50,000 from the TEM1 flight (scale 1:20.000). Land use maps of 2000, 2007, 2012 and 2015 were provided by the Lombardy Region and shared as part of the Infrastructure for Spatial Information in Lombardy (IIT) via the Geoportal (Lombardy Region Geoportal: http://www.cartografia.regione.lombardia.it/geoportale,

last access: 11 December 2017). The map of 2000 was obtained from the photo interpretation of aerial images of Flight IT2000, with a resolution of 1 m. While, the land use map of 2007 was realized by using colour and infrared orthophotos from Flight IT2007, with a resolution of 0.5 m. The maps of 2012 and 2015, which corresponded to the actual situation, were realized through the photo-interpretation of aerial photos realized by Agency for Disbursement in Agriculture (AGEA). The photo-interpretation was also supported by auxiliary data of Lombardy Region databases (e.g. Regional Agricultural Information System, Forest Types maps, map of the resident population, Archive of Integrated Activities production). The overall accuracies of maps obtained for Lombardy Region using this methodology was reported in Zaffaroni (2010) as approximately 95%. More detailed information about the method to realize these maps are available in Fasolini (2014)...". Furthermore, we moved the information about the inventory of affected road sectors in "The study area" section of the paper, adding this paragraph at pag. 5 lines 28-33 and pag. 6 lines 1-2 of the revised version of the manuscript: "...A detailed inventory map of the road sectors affected by shallow landslides in the study area was prepared and used as response variable of the model. The inventory map of the affected road traits include all the sectors hit by the shallow landslides occurred in the study area during 27–28 April 2009, March/April 2013 and 28 February–2 March 2014 rainfall events. For 2009 event, color aerial photographs at a resolution of 15 cm acquired immediately after the event were examined (Persichillo et al., 2017). For 2013 event, affected road traits were identified by visual interpretation of Pleiades satellite images with a resolution of less than 1 m (Persichillo et al., 2017). For 2014 event, slope failures and affected roads immediately after the event were detected through field surveys; the identified phenomena were mapped through a GPS tool, whose resolution is less than 2.5 m...". References are present in the References section of the revised version of the manuscript.

Comment 3 (ii) the discussion part must rewrote, at least the first 4 paragraphs.

Response to Comment 3 We modified the first four paragraphs of the "Discussions"

section, in order to clarify the exposed concepts. The modified parts were at pag. 15 lines 2-30 of the revised version of the manuscript. References are present in the References section of the revised version of the manuscript.

Comment 4 (iii) please change the Fig. 9, in order to make it representative according to the associated legend;

Response to Comment 4 We modified this figure for improving its comprehension.

Comment 5 and (iv) try to find other color for the roads (at least in the Figures 1, 3, and 9) for increasing the contrast and readability of the map.

Response to Comment 5 We modified the colors of road network in Figures 1 and 3 to improve its visualization.

Comment 6 Other minor problems you will find in the *.pdf file attached..

Response to Comment 6 We thank the Referee also for these suggestions. We considered all these minor revisions proposed by the Referee and we inserted them in th

Please also note the supplement to this comment:
https://www.nat-hazards-earth-syst-sci-discuss.net/nhess-2017-457/nhess-2017-457-AC3-supplement.pdf
* * *
**Fig. 1.** Distribution of some predictor variables considering to model road susceptibility: a) SL;
b) ASP; c) CURV; d) LEN; e) HEI; f) CA; g) CS; h) TWI; i) DIST.